# Fine mapping of MHC region in lung cancer highlights independent susceptibility loci by ethnicity

Aida Ferreiro-Iglesias et al.[#]

Lung cancer has several genetic associations identified within the major histocompatibility complex (MHC); although the basis for these associations remains elusive. Here, we analyze MHC genetic variation among 26,044 lung cancer patients and 20,836 controls densely genotyped across the MHC, using the Illumina Illumina OncoArray or Illumina 660W SNP microarray. We impute sequence variation in classical HLA genes, fine-map MHC associations for lung cancer risk with major histologies and compare results between ethnicities. Independent and novel associations within HLA genes are identified in Europeans including amino acids in the *HLA-B*0801* peptide binding groove and an independent *HLA-DQB1*06* loci group. In Asians, associations are driven by two independent HLA allele sets that both increase risk in *HLA-DQB1*0401* and *HLA-DRB1*0701*; the latter better represented by the amino acid Ala-104. These results implicate several HLA–tumor peptide interactions as the major MHC factor modulating lung cancer susceptibility.

Large scale genome wide association studies (GWAS) of lung cancer revealed different susceptibility loci across the main histological subtypes: adenocarcinoma (AD), squamous cell carcinoma (SCC), and small cell lung cancer (SCLC)[1,2]. This heterogeneity is particularly apparent for the Major Histocompatibility Complex (MHC), where associations have been observed specifically for SCC at 6p21.33 (rs3117582) in Europeans[3,4], and for AC at 6p21.32 (rs2395185) in Asians[5].

However, interpreting MHC associations is difficult as it is the most gene-dense region of the genome, is highly polymorphic, displays extensive linkage disequilibrium (LD), and genes are clustered by related functions[6]. In addition, this extreme variation is known to be shaped by population history and different infectious exposures[7,8]. Imputation of classical Human Leukocyte Antigen (HLA) at a four-digit resolution based on high-density SNP genotyping is an accurate and fast alternative to traditional HLA genotyping and permits the screening of large sample sets of different ethnicities[9,10].

Understanding the role of HLA in lung cancer is important, as it may help to elucidate the causal aetiology beyond the predominant role of smoking. Class I and II HLA molecules are known critical mediators in disease defense through presenting intra- or extra-cellular peptides on the cell surface in a form that can be recognized by the T cell receptors (TCR) and to then activate a specific T cell response[11]. To avoid immune-mediated elimination, cancer cells may lose their antigenicity due to different possibilities[12]. One is the immune selection of cancer cells which lack or mutate immunogenic tumor antigens[13]. Therefore, certain MHC alleles and polymorphisms may target particular tumor antigens, resulting in the observed disease-specific associations. A recent example is the association between an extended HLA haplotype (HLA-DRB1*1301–HLA-DQA1*0103–HLA-DQB1*0603) and protection against HPV associated oropharynx cancer[14], as well as cervical cancer[15]. These insights have the potential to inform tumor-specific immune responses and thus to be useful in developing immunotherapies. Tumor antigens can be derived from viral proteins, proteins encoded by cancer-germline genes, differentiation antigens and proteins arising from somatic mutations or gene rearrangements[16]. Identifying polymorphisms controlling expression of specific HLA molecules, affecting the peptide binding groove or the contact surface with the TCR may help to disentangle lung cancer MHC associations but also may provide new insights into cancer risk and possible immunotherapy targets[12].

To this end, we analyze genetic variation in two populations of European and Asian ancestry densely genotyped across the MHC in relation to lung cancer risk. Our results suggest that the genetic risk of the MHC region on lung cancer is different by population and by histology which points to different exposures or mechanisms interacting with HLA.

## Results

**Imputation of the HLA region.** After completion of imputation for the European and Asian series (18,924 cases/15,439 controls and 2324 cases/1656 controls, respectively) (Table 1), the final set of imputed variants used in association analysis were of high quality for Europeans, 92.5% of the variants had $R^2 \geq 0.9$, and 67.8% of the less common variants (MAF < 0.05) had $R^2 \geq 0.9$, but overall quality was less apparent for Asians as 60.5% of the total variants and 30.7% of the rare variants (MAF < 0.05) had $R^2 \geq 0.9$ for Asians. However common variants were well imputed (75% of the common variants and 98% of common HLA alleles (MAF > 0.05) had $R^2 \geq 0.9$). Given our sample size in Asians, our primary focus was on common variants for which the current imputation is satisfactory.

We also performed a laboratory validation of the variants identified by imputation using another genotyping platform, Affymetrix Axiom exome array[17], in a subset of 5742 individuals from the European series. A separate laboratory validation was not available for the Asian populations although we did replicate our results among never smokers in an independent series and we validated imputed HLA alleles using a completely different imputation algorithm and reference panel. Statistical re-imputation was done in a random 10% subset of the samples (3000 European samples and 1000 Asian samples from Oncoarray) using HIBAG[18]. The average concordance between imputed genotypes and their validation results either from an

---

**Table 1 Demographic characteristics of the participating studies after quality control filters**

| | European ancestry | | Asian ancestry | |
|---|---|---|---|---|
| | Case no. (%) | Control no. (%) | Case no. (%) | Control no. (%) |
| OncoArray studies- passed QC | 18,924 | 15,439 | 2324 | 1656 |
| Age | | | | |
| ≤50 | 2098 (11.1) | 2084 (13.5) | 242 (10.4) | 154 (9.3) |
| >50 | 16,801 (88.8) | 13,306 (86.2) | 2080 (89.5) | 1502 (90.7) |
| Missing | 25 (0.1) | 49 (0.3) | 2 (0.1) | 0 (0) |
| Sex | | | | |
| Male | 11,685 (61.7) | 9240 (59.8) | 1578 (67.9) | 1070 (64.6) |
| Female | 7236 (38.2) | 6196 (40.1) | 746 (32.1) | 586 (35.4) |
| Missing | 3 (0.02) | 3 (0.02) | 0 (0) | 0 (0) |
| Smoking status | | | | |
| Never | 1830 (9.7) | 4870 (31.5) | 815 (35.0) | 668 (40.3) |
| Ever | 16,682 (88.2) | 10,219 (66.2) | 1503 (64.7) | 983 (59.4) |
| Current | 9459 (50.0) | 4285 (27.8) | 986 (42.4) | 739 (44.6) |
| Missing | 412 (2.2) | 350 (2.3) | 6 (0.3) | 5 (0.3) |
| Histology | | | | |
| Adenocarcinoma | 7088 (37.5) | — | 1192 (51.3) | — |
| Squamous cell carcinoma | 4581 (24.2) | — | 641 (27.6) | — |
| Small cell carcinoma | 190 (10.5) | — | 94 (4.0) | — |
| Others[a] | 5265 (27.8) | | 397 (17.1) | |

[a]Large Cell, Bronchioloalveolar Carcinoma, Non-Small Cell Carcinoma, Carcinoids, Others or Missing

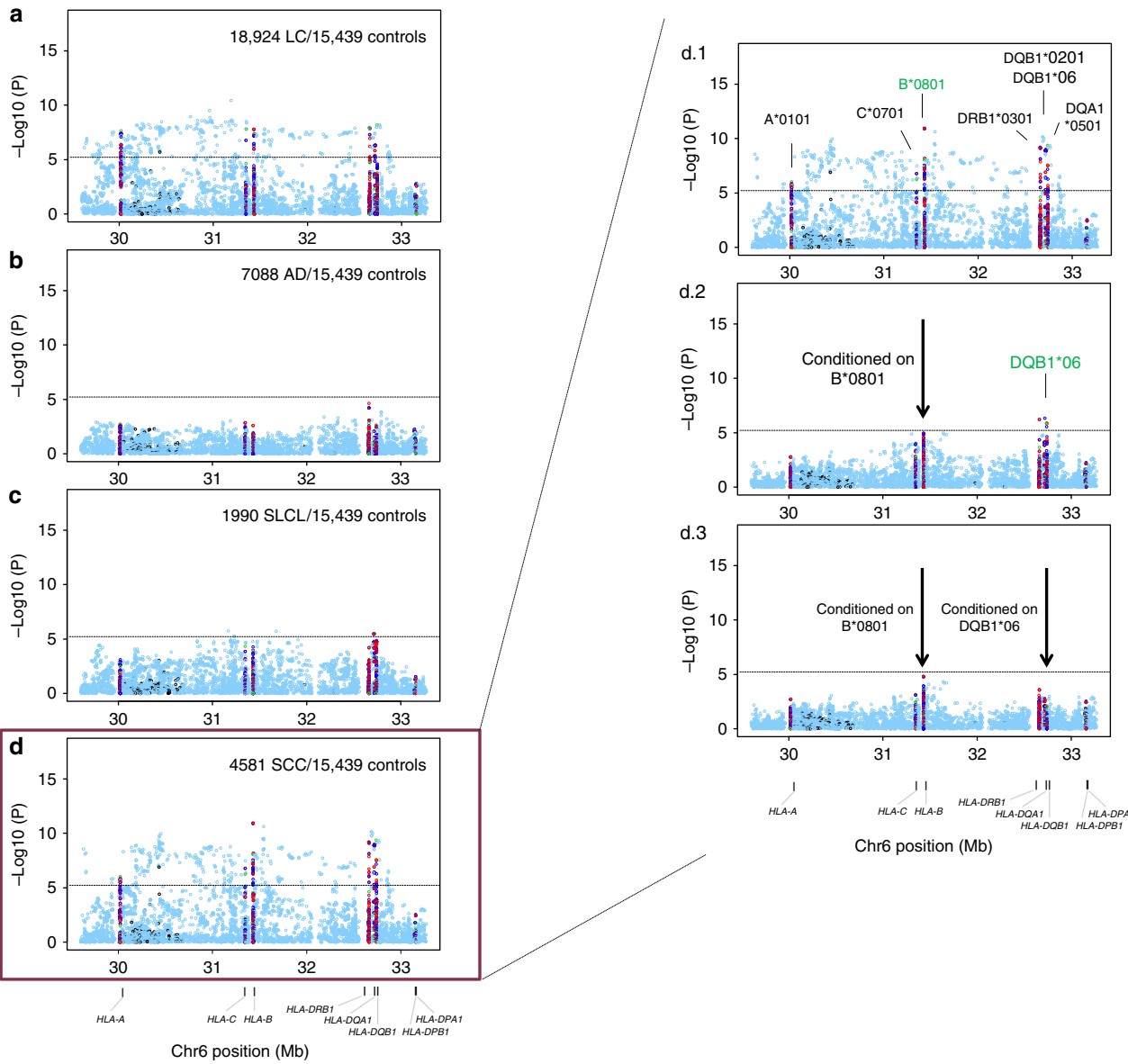

**Fig. 1** European regional association plots of variants in the MHC region and lung cancer overall and major histologies (a–d); plots of stepwise conditional analyses for squamous cell carcinoma (d.1–d.3). Each panel on the left shows the association plot for each unconditioned analysis **a** lung cancer overall, **b** adenocarcinoma, **c** small cell, **d** squamous cell carcinoma. The association for each locus used for conditioning is shown in green in each panel (d.1) unconditioned, (d.2) conditioned on *HLA-B*0801*, (d.3) conditioned on *HLA-B*0801* and *HLA-DQB1*06*. Detailed association results in Table 2 and Supplementary Fig. 1. Circles represent –log10 (P values) for each binary marker using the imputed allelic dosage (between 0 and 2). The dashed black horizontal lines represent the study-wide significant threshold of $P = 6 \times 10^{-6}$. The physical positions of HLA genes on chromosome 6 are shown at the bottom. The color of the circles indicates the type of marker; light blue—SNPs outside HLA genes, green—classical HLA alleles and red—amino acid polymorphisms of the HLA genes; deep blue—SNP within HLA genes)

alternative genotyping platform or re-imputation was >95% for all variants (Supplementary Table 10 and 11).

The best-guess genotype dosages ($R^2 \geq 0.7$) for two- and four-digit classical alleles, as well as amino acid polymorphisms and SNPs in the MHC region were tested for association with overall lung cancer and subtypes in European (Fig. 1 and Table 2) and Asian ancestries (Fig. 2 and Table 3).

**HLA region association analyses.** Multiple association signals were detected in Europeans within class I and class II regions for lung cancer overall (Fig. 1a). Stratified analyses showed that these associations were restricted to SCC (Fig. 1d) with little evidence of associations for AD (Fig. 1b) and SCLC (Fig. 1c). In contrast,

MHC associations in Asians were less abundant in the overall analyses (Fig. 2a), and after stratifying by major histologies we observed the top association signals for lung AD (Fig. 2b). Therefore, subsequent explanations for Europeans and Asians will refer to SCC and AD results, respectively.

In Europeans, the most significant allele was the GT allele of a multiple nucleotide variation at Chr6: 31431982–31431983 (NCBI build 36), a complex variant affecting the first and second nucleotides of *HLA-B* codon 163 (OR = 1.20, $p = 1.30 \times 10^{-11}$; Fig. 1d. 1). This allele codes for Thr-163 or Glu-163 in *HLA-B*. Thus, the strongest MHC signal mapped to an amino acid change resulting in a Thr-163 in *HLA-B*, which is part of the sequence of the four-digit allele *HLA-B *0801* that showed an equivalent association (Table 2).

**Table 2 Top associations of the HLA alleles with squamous cell carcinoma of European ancestry**

| Locus | HLA variant | Frequency | | Unconditional analysis[a] | | Model including AH8.1 + HLA-DQB1*06 | |
|---|---|---|---|---|---|---|---|
| | | Controls[c] | Cases[c] | OR (95% CI) | P value | OR (95% CI) | P value |
| AH 8.1 | A1-B8-DR3-DQ2 | 0.06 | 0.07 | 1.30 (1.18-1.42) | $4.78 \times 10^{-08}$ | 1.24 (1.14-1.37) | $2.13 \times 10^{-06}$ |
| HLA-A | 01:01 | 0.15 | 0.17 | 1.18 (1.10-1.25) | $9.36 \times 10^{-07}$ | | |
| HLA-C | 07:01 | 0.15 | 0.17 | 1.17 (1.10-1.25) | $5.53 \times 10^{-07}$ | | |
| HLA-B | 08:01 | 0.10 | 0.12 | 1.25 (1.16-1.34) | $9.01 \times 10^{-09}$ | | |
| HLA-DRB1 | 03:01 | 0.11 | 0.14 | 1.25 (1.16-1.34) | $6.38 \times 10^{-10}$ | | |
| HLA-DQA1 | 05:01 | 0.20 | 0.29 | 1.18 (1.12-1.24) | $1.33 \times 10^{-09}$ | | |
| HLA-DQB1 | 02:01 | 0.11 | 0.14 | 1.25 (1.17-1.34) | $4.45 \times 10^{-10}$ | | |
| HLA-DQB1 | 06 global[b] | 0.24 | 0.21 | 0.85 (0.80-0.90) | $3.05 \times 10^{-08}$ | 0.86 (0.78-0.91) | $9.96 \times 10^{-08}$ |
| | 06:01 | 0.01 | 0.01 | 0.87 (0.67-1.12) | 0.3 | | |
| | 06:02 | 0.12 | 0.10 | 0.90 (0.83-0.97) | 0.007 | | |
| | 06:03 | 0.07 | 0.06 | 0.84 (0.76-0.93) | $5.06 \times 10^{-04}$ | | |
| | 06:04 | 0.04 | 0.03 | 0.86 (0.75-0.99) | 0.03 | | |
| | 06:09 | 0.009 | 0.007 | 0.80 (0.61-1.06) | 0.1 | | |

AH 8.1 ancestral haplotype 8.1, HLA human leucocyte antigen, OR odds ratio, 95% CI confidence interval
[a]Obtained from multivariate unconditional logistic regression assuming an additive genetic model with sex and principal components as covariates
[b]Classical two-digit allele accounting for the four digit alleles found (*0601,*0602,*0603,*0604,*0609)
[c] Number of samples included in the analysis: 4,581 cases and 15,439 controls
The study-wide significant threshold was $P = 6 \times 10^{-6}$ (Bonferroni correction)

Other signals including HLA alleles were also detected across the MHC (Table 2; Fig. 1d. 1). A stepwise conditional logistic regression analysis was performed to identify variants that independently influence lung SCC susceptibility in Europeans. First, conditioning on HLA-B *0801, HLA-DQB1 *06 remained as the highest peak (OR = 0.85, $p = 3.05 \times 10^{-8}$; Fig. 1d. 2), while other variants were not significant. In a second step, we included both HLA-B *0801 and HLA-DQB1*06 as covariates in the analysis and did not detect any remaining independent signals (Fig. 1d. 3). In Asians from Oncoarray, we also observed several associations along class I and class II (Fig. 2b. 1). The A allele of rs3129860 located in an intergenic region within HLA class II was the top associated marker. As this SNP was highly correlated with HLA-DQB1*0401 ($r^2 = 0.75$) and apparently not showing a potential regulatory function, we used this allele in subsequent steps. After conditioning on classical HLA-DQB1*0401 allele, HLA-DRB1*0701 remained as the most significant signal (Fig. 2b. 2). When we controlled for both alleles, rs2256919 (an HLA-A intronic variant) remained associated (Figure 2b. 3). Finally, after controlling for HLA-DQB1*0401, HLA-DRB1*0701 and rs2256919 no additional variants remained associated with AD risk with a significance threshold (conditioned $P > 6 \times 10^{-6}$; Figure 2b. 4).

Results of the stratified analyses by histology and by smoking status for each of the independent variants found in Europeans and Asians are summarized in Supplementary Fig. 1. We observed no significant differences by smoking status for any of the analyzed variants. However, AH8.1 showed a risk effect in the overall analysis as well as in the smoking groups, but a protective effect (OR = 0.88) in the never-smoking group which involved a significant heterogeneity ($p_{het} = 0.006$, Supplementary Fig. 1a). We wanted to explore further the impact of this effect in the overall analysis by adding smoking as a covariate for AH8.1 and related markers, but also for the rest of associations in both ethnithities. As can be seen in Supplementary Tables 8 and 9, results are extremely similar to the original results in Tables 2 and 3, indicating that adjustment by smoking makes little difference, as expected. In consequence, we can consider all the associated markers practically independent on smoking status.

No significant result or trend was detected for the European associated variants when we checked their association in Asians or vice versa. A possible reason for certain alleles highlighted in our analysis might be their different frequencies and meaning in European and Asian populations (Tables 2 and 3; Supplementary Tables 2 and 3). For example, European hits like class I alleles and DQB1*06 are very rare in Asians (<2%) (Supplementary Table 2) but common in Europeans (>10%) (Table 2). However, other alleles showed similar frequencies (>10%) in both populations even they were associated just in one. Statistical power calculations (Supplementary Tables 12 and 13) showed that the Asian set sample size was insufficient for detecting some European hits but not the opposite. This suggests ethnic-specific effects due to population history although we cannot discard the same effects in squamous cell carcinoma in both populations. However, it seems that HLA is not playing a role in adenocarcinoma in Europeans, unless the effects are hidden in very rare variants

**Haplotype analysis.** Because of the broad LD of the region, we wanted to assess the physical genetic boundaries of these associations on the basis of haplotype patterns. Supplementary Figs. 2 and 3 include an overview of the haplotypes detected in cases and controls of European and Asian ancestry, respectively. In Europeans, the most frequent haplotype and the only one associated with any of the tested outcomes was the ancestral haplotype 8.1 (AH8.1) (~6 %). AH8.1 contains class I and class II HLA alleles (A*0101 - B*0801 - C*0701 - DRB1*0301 - DQB1*0201 - DQA1*0501) and showed increased risk for SCC in Europeans (Table 2; Supplementary Fig. 1a; Supplementary Fig. 2). Conditional analyses considering AH8.1 and HLA-DQB1 *06 in the same model, revealed the independence of both effects (Table 2). However, no significant differences were detected in haplotype frequencies for any of the outcomes for Asians.

**Analysis of polymorphic amino acid positions and best model selection.** Although we identified a HLA haplotype and a 2-digit allele group independently associated with SCC in Europeans we aimed to answer the more refined question as to whether the association within the MHC resides with HLA alleles only, amino acids only, or a combination of both HLA alleles and amino acids. To answer this, we searched for the best combination of amino acids and/or HLA alleles that explained the HLA haplotype

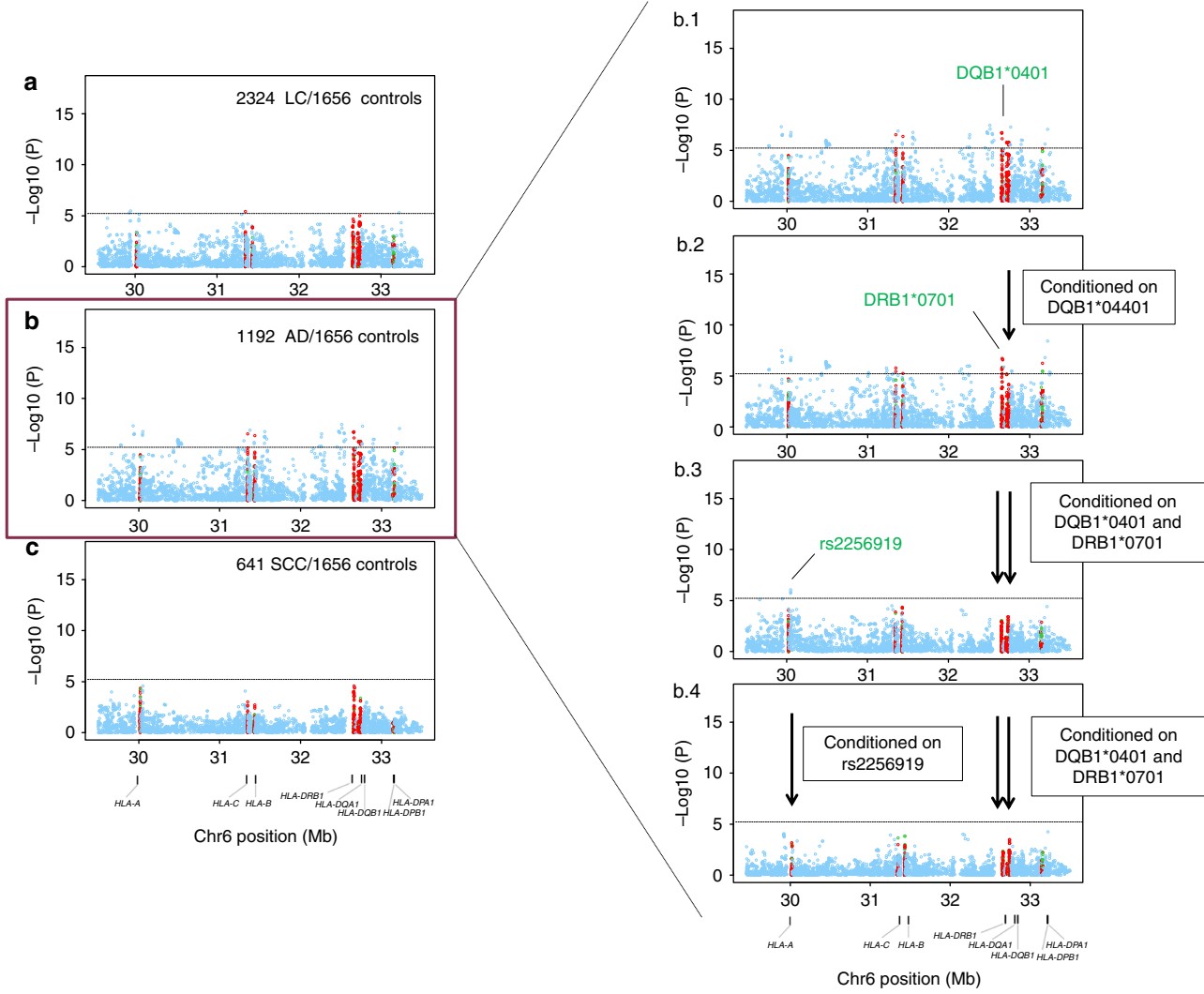

**Fig. 2** Asian regional association plots of variants in the MHC region and lung cancer overall and major histologies (**a–c**); plots of stepwise conditional analyses for lung adenocarcinoma (b. 1–b. 4). Each panel on the left shows the association plot for each analysis **a** lung cancer overall, **b** adenocarcinoma, **c** squamous cell carcinoma. The association for each locus used for conditioning is shown in green in each panel on the right (b. 1) unconditioned, (b. 2) conditioned on *HLA-DQB1*0401*, (b. 3) conditioned on *HLA-DQB1*0401* and *HLA-DRB1*0701*, (b. 4) conditioned on *HLA-DQB1*0401, HLA-DRB1*0701* and rs2256919 (SNP within *HLA-A*). Detailed association results in Table 3 and Supplementary Figure 1. Circles represent –log10 (P values) for each binary marker using the imputed allelic dosage (between 0 and 2). The dashed black horizontal lines represent the study-wide significant threshold of $P = 6 \times 10^{-6}$. The physical positions of HLA genes on chromosome 6 are shown at the bottom. The color of the circles indicates the type of marker; light blue—SNPs, green—classical HLA alleles, and red—amino acid polymorphisms of the HLA genes

independently on the 2-digit allele group associated with SCC. Our model selection criterion was the Bayesian Information Criterion (BIC) since this has a heavy penalty for variable inclusion. However, the results did not differ from those obtained with the Akaike Information Criterion (AIC), a model-choice that uses a weaker penalty (when sample sizes are large) than does the BIC for the inclusion of variables.

As starting point, we ran an unconditional logistic regression (see Materials and methods) for each of the amino acids of the HLA genes and then searched for the best model from individual HLA allele or amino acid using the BIC criterion. We observed the most significant associations at HLA-B, HLA-DRB1 and HLA-DQB1 genes in Europeans.

HLA-B and HLA-DRB1 amino acids effects at Asp-9 (tagging HLA-B*0801), Thr-163 and Asp-156 in B and Lys-71, Arg-74 and Tyr-26 in DRB1 are located in protein binding grooves and part of the AH8.1 HLA alleles, which adds functional relevance to our

data (Table 2, Fig. 3, Supplementary Table 4). Thr-163 and Arg-74 individual models were the ones that best explained the data in HLA-B and HLA-DRB1, respectively, as they presented the lowest BIC value.

Amino acids associated in HLA-DQB1 were shared exclusively by HLA-DQB1*06 molecules, i.e., not present in any other allele in HLA-DQB1 detected in the analyzed samples. As these were not located in protein binding grooves or any potential regulatory function and both two-digit allele and amino acid individual models showed similar support, we used HLA-DQB1*06 in subsequent analysis (Supplementary Table 4).

The following variables were consistently included in the combined models: B-Asp-9, B-Thr-163, B-Asp-156, *HLA-B*0801*, DRB1-Lys-71, DRB1-Arg-74 and DRB1-Tyr-26, *HLA-DRB1*0301* and *HLA-DQB1*06*. Table 4 displays the combined models with the best fit to the data from this search using a stepwise conditional logistic regression approach. There is some

**Table 3 Top associations of the HLA alleles with adenocarcinoma of Asian ancestry**

| Locus | Variant | Frequency | | Unconditional analysis[a] | | Model including *HLA-DQB1 *0401* + *HLA-DRB1*0701* + rs2256919 | |
|---|---|---|---|---|---|---|---|
| | | Controls[b] | Cases[b] | OR (95% CI) | P value | OR (95%CI) | P value |
| HLA-DQB1 | *04:01* | 0.06 | 0.09 | 1.67 (1.35-2.05) | $1.59 \times 10^{-06}$ | 1.73 (1.41-2.14) | $2.85 \times 10^{-07}$ |
| HLA-DRB1 | *07:01* | 0.05 | 0.09 | 1.62 (1.31-2.01) | $5.48 \times 10^{-06}$ | 1.63 (1.32-2.03) | $5.34 \times 10^{-06}$ |
| HLA_A (intronic) | rs2256919 | 0.44 | 0.38 | 0.75 (0.67-0.83) | $1.75 \times 10^{-07}$ | 0.76 (0.68-0.85) | $8.92 \times 10^{-07}$ |

*HLA* human leucocyte antigen, *OR* odds ratio, *95% CI* confidence interval
[a] Obtained from multivariate unconditional logistic regression assuming an additive genetic model with sex and principal components as covariates
[b] Number of samples included in the analysis: 1192 cases and 1656 controls
The study-wide significant threshold was $P = 6 \times 10^{-6}$ (Bonferroni correction)

uncertainty as to the best model that relates to whether we find evidence of a three signal model at B-Thr-163, DRB1-Arg-74 and *HLA-DQB1*06* (model B), or just two signals at B-Thr-163 and *HLA-DQB1*06* (model C). However, DRB1-Arg-74 does not seem to be an independent effect on B-Thr-163 since the effect was not significant and weaker within model B (OR = 1.11; $p = 8.87 \times 10^{-3}$), whereas the effect was stronger and significant in model D (OR = 1.19; $p = 13 \times 10^{-7}$) and in the individual model (OR = 1.25; $p = 13 \times 10^{-8}$). Accordingly, model C was considered the best model fitting the data. This implies that the amino acid Thr-163 in *HLA-B*0801* is sufficient to explain the risk in the AH8.1 haplotype for lung SCC in Europeans (Table 2; Table 4; Supplementary Table 4; Fig. 3a).

In Asians, the most relevant amino acid positions for AD risk were Ala-104 and Glu-98 (in tight LD, $r^2 = 1$) in HLA-DRB1 and Leu-23 in HLA-DQB1 (Supplementary Table 5); their individual models fitted the data as well as *HLA-DRB1*0701* and *HLA-DQB1*0401* alleles (Supplementary Table 5). However, when the amino acids were included instead of alleles the model had no support (Supplementary Table 6). In addition, these amino acids were not located in the binding groove of the corresponding molecules, thus a priori there is no functional relevance behind or motivation for considering these amino acid positions over the alleles as the best variants to explain the data.

**Asian replication and meta-analyses.** In order to give more reliability to Asian results we included an additional set of 8,537 samples obtained from published GWAs[5]. This is a multicenter collection of non-smoking women that serves as a replication collection as far as none of the hits were dependent on smoking status or gender in the discovery phase (Supplementary Fig. 1, Supplementary Tables 8 and 9). Therefore, we considered it reasonable to meta-analyze both Asian datasets and then extend our comparison between ethnicities.

In Asians from Lan et al., we did not observed any associations in the overall analysis (Supplementary Fig. 4a). As in the first phase, after stratifying by major histologies we observed the top association signals for lung AD (Supplementary Fig. 4b) and none for SCC (Supplementary Fig. 4c).The T allele of rs2856688 located in an intergenic region within HLA class II was the top associated marker. As this SNP was highly correlated with an amino acid change resulting in a Ala-104 in HLA-DRB1*0701 and HLA-DRB1*0401 (r2 = 0.95), we used this amino acid in subsequent steps. After conditioning on classical HLA-DQB1 Ala-104 no additional variants remained associated with AD risk with a significance threshold (conditioned $P > 6 \times 10^{-6}$; Supplementary Figure 4b. 2).

To gain insights into modest signals of association, we combined the imputed results of the Asian Oncoarray and Lan et al. using a random-effect meta-analysis approach. Results from top Asian associated variants are displayed in Table 5. There is a

modest yet significant heterogeneity between the studies, with results from Oncoarray present stronger effects, even for the variant originally reported in Lan et al. (Ala104/Ser104 can be consider a surrogate ($r^2 = 0.99$) of this variant, rs2395185). The two main HLA effects detected in Asian Oncoarray analysis, in HLA-DQB1 and HLA DRB1, exceed the threshold of significant in the combined analysis showing a consistence with the results of the first phase. The HLA-A intronic SNP rs2256919 was not however replicated.

**Discussion**

We have performed a comprehensive association analysis of HLA alleles, SNPs and polymorphic amino acid sites that identified several independent effects and their most likely causal variants that lead to the association of the MHC with lung cancer in Europeans and in Asians. In addition, our results suggest that the genetic risk of the MHC region on lung cancer is different between Asian and European populations.

Regarding the European data, the key findings are the AH8.1 haplotype association with SCC and a second independent signal involving several *HLA-DQB1*06* alleles. The main signal driving the AH8.1 effect mapped to the *HLA-B*0801* amino acid position 163 located in the protein binding groove.

In relation with previous findings, our analyses refined the association within *BAG6/BAT3* susceptibility region encompassing rs3117582. This variant is in high LD ($r^2 = 0.76$) with *HLAB*0801*, therefore within AH8.1, and was reported as associated with SCC in the largest lung cancer GWAs published to date[3,4]. It was not certain whether these genes themselves or others in linkage disequilibrium (LD) were primarily responsible for these findings. Here, we demonstrate that the presence of a threonine in the amino acid position 163 of *HLA-B*0801* accounts for the main part of this effect as it fitted the data in the conditional model and it has biological significance. However, alleles within AH8.1 are in strong (but incomplete) LD[19]. For that reason we can't exclude additional loci within the haplotype supporting or contributing additively to risk as probably is happening with the amino acid position 74 in *HLA-DRB1*0301*, also located in the protein binding groove.

AH8.1 is a well-known Caucasian haplotype commonly associated with immune-mediated diseases[19–21]. Typically defined by A*0101 - B*0801 - C*0701 - DRB1*0301 - DQB1*0201 - DQA1*0501, it is the second longest haplotype identified within the human genome. As it is common and stable in Caucasians, it might have been advantageous in past environments, resistant to recombination and positively selected over the time[19]. Similar genetic associations reported for several autoimmune diseases suggest common mechanisms of immune dysregulation[21,22]. The amino acid position 163 in HLA-B is a functionally important residue for TCR recognition and is also associated with TCR

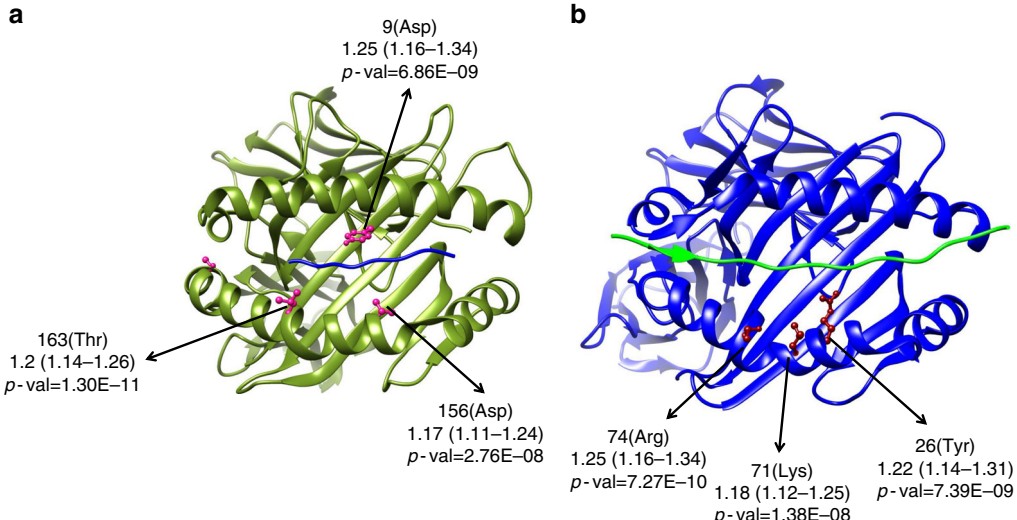

**Fig. 3** Three-dimensional ribbon models for the HLA-B (**a**) and HLA-DR (**b**) proteins. These structures are based on Protein Data Bank entries 2bvp and 3pdo, respectively, with a direct view of the peptide-binding groove. Key amino acid positions identified by unconditioned association analyses (squamous cell carcinoma of European ancestry) are highlighted. This figure was prepared using UCSF Chimera (see URL)

expression[23]. Future functional analyses are needed to confirm the impact of our findings.

The *HLA-DQB1*06* allele group showed a protective effect independent of AH8.1. This association is based in two amino acid positions (125 and 87) common to *DQB1*06* 4-digit alleles (*HLA-DQB1*0601*, *HLA-DQB1*0602*, *HLA-DQB1*0603*, *HLA-DQB1*0604* and *HLA-DQB1*0609*) but not present in any other DQB1 alleles detected here. It is possible that one of them is driving the effect and the other is just in tight LD ($R^2 = 0.78$). Alternatively, there may be a joint effect involving both amino acids, driven by combined selection. This is plausible given the important role of natural selection in the MHC[22,24], even if these are not in a clear functional physical location. The allele showing the strongest association trend within DQB1*06 group is *HLA-DQB1*0603*, part of the HLA class II haplotype *HLA-DRB1*1301–HLA-DQA1*0103–HLA-DQB1*0603* (in tight LD in Europeans, $r^2 > 0.9$) and associated with head and neck[14] and cervical cancer[15], both squamous cell carcinomas linked to HPV infection. For lung SCC this haplotype showed a trend of association with the same protective effect as in the HPV associated cancers (Supplementary Table 14).

In contrast, in the Asians Oncoarray analysis we only observed associations between HLA class II alleles and risk of lung adenocarcinoma. We observed two independent HLA allele effects increasing risk, *HLA-DQB1*0401* and *HLA-DRB1*0701*, as well as a protective intronic SNP in HLA-A (rs2256919). A potential limitation of our study is the relatively modest sample size of the Asian data set, compared with the European population. However, in the replication phase, we observed two of the three independent HLA allele effects that were described in the first phase. One is represented by the four-digit allele *HLA-DQB1*0401* as in the discovery, and the other is better explained by the association of Ala-104/Glu-98/Gln-10, part of *HLA-DRB1*0701* and *HLA-DRB1*0401* protein sequence. This is the same effect as the one described by Lan et al. in the intergenic region in 6p21.32 (rs2395185) as all, SNP and amino acids changes, are in tight LD ($r^2 = 0.99$). We could not confirm the association of the intronic SNP in HLA-A (rs2256919).

We also observed no overlap between Asian and European hits. While we cannot rule out the possibility that some of the observed differences could be due to the lack of power in the Asian collection, it is possible that differences in the spectrum of antigens in the two populations might introduce changes in which alleles might play the important role in disease susceptibility within each population. However, prevalence of never smokers is another important difference between the Asian and European data sets. While we consider it beyond doubt that Asian HLA associations are independent on smoking, especially as we have replicated our initial findings in nonsmoking Asian women, is true that the proportion of non-smokers in Asian AD (469/1192; 39%) is higher than in European AD (1004/7088; 14%) and this could represent a problem if there is a dependency on smoking. Despite this, we did not see the association in European AD non-smokers even though the absolute number of never-smokers among European AC cases was greater than among Asians, and we had enough power to detect the effects shown in Asian AD.

In summary, this evidence suggests a different aetiological role for the MHC by population and by histology which points to different exposures or mechanisms interacting with HLA. These differences are clear for AD; however differences between ethnicities for SCC still need confirmation. Taken together, our findings enhance the role of HLA variants in the immune pathogenesis of lung cancer, and may also have implications for cancer immunotherapies targeting the HLA antigen presentation pathway[16,25,26].

## Methods

**Sample collections and genotypes**. Participants were of either European or Asian ethnicity and came from 30 case-control studies that were included in the OncoArray study, and part of the Transdisciplinary Research of Cancer in Lung of the International Lung Cancer Consortium (TRICL-ILCCO) and the Lung Cancer Cohort Consortium (LC3) (Supplementary Table 1). After quality control, the European ethnicity series included 18,924 lung cancer cases and 15,439 controls, whilst the Asians series comprised 2324 cases and 1656 controls (Table 1). Informed consent was obtained for all participants, and studies were approved by respective institutional review boards.

For all samples, we had access to genome-wide SNP data at individual level from OncoArray genotyping platform[27]. The Illumina OncoArray custom was designed for cancer studies by the OncoArray Consortium, part of the Genetic Associations and Mechanisms in Oncology (GAME-ON) Network that includes fine-mapping of common cancer susceptibility loci with special emphasis on HLA region among others. Oncoarray genotyping and genotype quality controls procedure were done in the context of a large lung cancer genome-wide study using the OncoArray platform[4]. Briefly, genotype calls and quality control filters were made by the Dartmouth team in GenomeStudio software (Illumina) using a standardized cluster file for OncoArray. Standard quality control procedures were

**Table 4 HLA alleles and amino acids contained in the best models obtained for SCC in Europeans judged by the BIC criterion**

| Model | Locus | OR (95%CI) | P value | BIC | BIC dif[i] |
|---|---|---|---|---|---|
| A | | | | 21605 | +14 |
| AH8.1 + HLA-DQB1 *06 | | | | | |
| | AH8.1 | 1.24 (1.14-1.37) | $2.13 \times 10^{-6}$ | | |
| | HLA-DQB1 *06 | 0.86 (0.78-0.91) | $9.96 \times 10^{-8}$ | | |
| B | | | | 21594 | + 3 |
| B-Thr163 + DRB1-Arg74 + HLA-DQB1 *06 | | | | | |
| | B(163) | 1.13 (1.07-1.20) | $1.68 \times 10^{-5}$ | | |
| | DRB1(74) | 1.11 (1.03-1.20) | $8.87 \times 10^{-3}$ | | |
| | HLA-DQB1 *06 | 0.87 (0.82-0.92) | $1.15 \times 10^{-6}$ | | |
| C | | | | 21591 | 0 |
| B-Thr163 + HLA-DQB1 *06 | | | | | |
| | B(163) | 1.17 (1.11-1.23) | $1.83 \times 10^{-9}$ | | |
| | HLA-DQB1 *06 | 0.86 (0.81-0.91) | $8.47 \times 10^{-8}$ | | |
| D | | | | 21603 | +12 |
| DRB1-Arg74 + HLA-DQB1 *06 | | | | | |
| | DRB1(74) | 1.19 (1.11-1.28) | $6.59 \times 10^{-7}$ | | |
| | HLA-DQB1 *06 | 0.87 (0.82-0.91) | $8.10 \times 10^{-7}$ | | |

*BIC* Bayesian information criterion, *HLA* human leukocyte antigen
[c] Number of samples included in the analyses: 4581 cases and 15,439 controls
[i] models having their BIC within: + 1–2 of the minimum have substantial support; + 4–7 of the minimum have considerably less support; >10 above the minimum have either essentially no support

used to exclude underperforming genotyping assays (judged by success rate, genotype distributions deviated from that expected by Hardy Weinberg equilibrium). Additionally, individuals with low genotyping success rate (<95%) and individuals with a genetically inferred gender by X and Y markers did not match that reported gender, or had excess identity by descent sharing relative to other samples, were excluded. Principal components analysis (PCA) was done using FlashPCA[3] and identified 10 and 3 significant eigenvectors on the European and the Asian dataset, respectively, that were used as covariates in the association analyses. PCA plots are displayed in Supplementary Fig. 5. Furthermore, generalized linear models showed that of the eigenvectors were significantly associated with the study of recruitment either in Asian or in European series (p-value: $< 10^{-16}$). Therefore, we have used eigenvectors in lieu of study to better adjust for genetic/ethnic origin.

An additional Asian series including 4962 lung cancer cases and 3845 controls was downloaded from DbGaP[5] (dbGaP reference: phs000716.v1.p1) and was used as a replication collection of the initial Asian results. Genome-wide data came from two sources in this study but we only used genotypes from the Illumina 660W SNP microarray as this was the source of the majority of the samples and of the same Asian origin. On top of the quality controls done[5], 270 duplicated samples were excluded. To account for potential population stratification, we performed PCA in EIGENSTRAT[28] using approximately 10,000 common markers in low LD ($r^2 < 0.004$, minor allele frequency (MAF) > 0.05). Subsequently, we derived the 7 significant eigenvectors to adjust association analyses. PCA plots are displayed in Supplementary Fig. 5. Supplementary Table 7 shows demographic characteristics of the final replication set 4,741 lung cancer cases and 3,796 controls, after quality controls.

**Imputation of the HLA system**. Taking advantage of the high OncoArray SNP coverage in the HLA region, we used the genotyping data from 25 to 35 Kb at chromosome 6 (NCBI build 37) obtained to impute classical two and four digit HLA alleles and amino acid polymorphisms of the HLA genes along with the SNPs that were not directly genotyped.

For Europeans, we imputed HLA variants using the reference data collected by the Type 1 Diabetes Genetics Consortium (T1DGC)[6,29]. This is a panel composed of 5,225 individuals of European origin with genotyping data for 8,534 SNPs and 424 classical HLA alleles of class I (*HLA-A*, *HLA-B* and *HLA-C*) and class II (*HLA-DRB1*, *HLA-DQB1*, *HLA-DQA1*, *HLA-DPB1* and *HLA-DPA1*) genes. For Asians, we used the Pan-Asian reference panel[30,31] which contains genotype data for 8,245 SNPs and 273 classical HLA alleles tagging the entire MHC, as described above, in 530 unrelated individuals of Asian descent.

Imputation process was performed with the SNP2HLA v1.0.3 package using Beagle software[32,33] (http://www.broadinstitute.org/mpg/snp2hla). It was conducted separately for each ethnic group and we imputed cases and controls together in randomized groups of approximately 1000 individuals. We applied post-imputation QC criteria of $R^2 < 0.3$ for excluding variants in the association analysis.

**Imputation validation**. To confirm imputed HLA alleles, we re-imputed[34] a random 10% subset of the samples (3000 European samples and 1000 Asian samples from Oncoarray) using HIBAG[18],which employs another pre-trained referenced panel and a different statistical method based on multiple expectation-

maximization-based classifiers to estimate the likelihood of HLA alleles (http://www.biostat.washington.edu/~bsweir/HIBAG/). Previous studies comparing the accuracy of HIBAG and SNP2HLA (among others imputation methods) to sequence data, concluded that they are the most robust programs with respect to maintaining accuracy[35,36]. In order to assess the accuracy of the imputation, we compared the imputed data for HLA alleles of class I (*HLA-A*, *HLA-B* and *HLA-C*) and class II (*HLA-DRB1*, *HLA-DQB1*, *HLA-DQA1*), of those HLA genotypes obtained in the same individuals with the two methods described above. Two parameters were considered for this comparison 1) a correlation coefficient (r), which is a measure of the reliability of the frequencies, and 2) the accuracy, to establish reproducibility of the typing in each individual, as described elsewhere[37].

We also performed a laboratory validation genotyping to confirm array imputed dosages of the associated variants using another genotyping platform Affymetrix Axiom exome array[17] in a subset of 5,742 individuals from the European series. We considered the associated variants that achieved the study significance in Europeans. For these loci, the sentinel variant (or correlated proxy variant) was also genotyped on the custom Affymetrix Axiom exome array. We therefore considered the concordance between the OncoArray genotypes and the Affymetric array for these variants in the 5,742 individuals where genotyping was available for both platforms. Supplementary Table 11 describes the satisfactory concordance between OncoArray (imputation) genotypes and the validation genotypes.

**Analysis across the MHC region**. All the tested HLA variants were defined as binary markers as follow: for biallelic SNPs, classical HLA alleles and binary amino acid positions, the effect allele or variant was the minor allele, the presence of the HLA allele or the presence of the less frequent amino acid. For multi-allelic amino acid positions we defined composite markers where each possible individual allele and combination of alleles was tested for association.

To test for association within the HLA region and given the ancestral variation of our study, we evaluated associations separately for each ethnicity (European and Asian). For each marker we used dosages, which take uncertainty in imputation into account, in multivariate unconditional logistic regression models under a log-additive genetic model controlling for sex and principal components (as was described above). We assessed the association between the described variants and lung cancer risk, as well as predominant histological types and smoking behaviour. We set a study-wide significance threshold of $P = 6 \times 10^{-6}$, on the basis of the highest total number of genotyped SNPs, as well as imputed SNPs, HLA alleles and amino acid variants that were included in the analyses after passing QC for Europeans (0.05/8,291) and Asians (0.05/5,504). The number of markers that passed QC in the Asian series is less but we maintained in the Asian analysis the more conservative p-value cut-off adopted for Europeans. This correction far exceed number of independent test expected within this region given its high linkage disequilibrium, and it can be considered a very conservative p-value.

To assess whether there were independent effects outside of the main associated loci, we used a conditional additive logistic regression approach to test all markers across the MHC. In order to explore whether any of the significant variant identified in this process have any potential regulatory function if they are not in a critical location, we used the online tool HaploReg v4.1[38] to confirm location of each SNP in relation to annotated protein-coding genes and/or non-coding regulatory elements. As a first step, we included *HLA* alleles as covariates when they or their tag-SNPs (not annotated as functional) appeared as the best associated

**Table 5 Meta-analyses of top associated HLA variants with adenocarcinoma of Asian ancestry**

| Locus | Variant | Study | N | Frequency | Unconditional analysis | | Cochran's Q |
|---|---|---|---|---|---|---|---|
| | | | Controls/Cases | Controls/Cases | OR (95%CI) | P value | P_het |
| HLA-DQB1 | 04:01 | Oncoarray[a] | 1656 / 1192 | 0.06 / 0.09 | 1.67 (1.35-2.05) | $1.59 \times 10^{-06}$ | |
| | | Lan et al.[b] | 3741 / 3469 | 0.06 / 0.07 | 1.22 (1.07-1.40) | $4.69 \times 10^{-04}$ | |
| | | Combined[c] | 5129 / 4576 | 0.06 / 0.08 | 1.34 (1.19-1.51) | $4.88 \times 10^{-07}$ | 0.02 |
| HLA-DRB1 | 07:01 | Oncoarray[a] | 1656 / 1192 | 0.05 / 0.09 | 1.62 (1.31-2.01) | $5.48 \times 10^{-06}$ | |
| | | Lan et al.[b] | 3741 / 3469 | 0.06 / 0.07 | 1.10 (0.96-1.26) | 0.15 | |
| | | Combined[c] | 5129 / 4576 | 0.06 / 0.08 | 1.23 (1.10-1.38) | $3.66 \times 10^{-04}$ | 0.003 |
| | Ala104/Ser104[d] | Oncoarray[a] | 1656 / 1192 | 0.36 / 0.43 | 1.34 (1.20-1.50) | $5.48 \times 10^{-06}$ | |
| | | Lan et al.[b] | 3741 / 3469 | 0.35 / 0.39 | 1.16 (1.08-1.24) | $5.11 \times 10^{-06}$ | |
| | | Combined[c] | 5129 / 4576 | 0.35 / 0.40 | 1.20 (1.14-1.28) | $5.21 \times 10^{-10}$ | 0.03 |
| HLA_A (intronic) | rs2256919 | Oncoarray[a] | 1656 / 1192 | 0.44 / 0.38 | 0.75 (0.67-0.83) | $1.75 \times 10^{-07}$ | |
| | | Lan et al.[b] | 3741 / 3469 | 0.43 / 0.42 | 0.96 (0.90-1.03) | 0.26 | |
| | | Combined[c] | 5129 / 4576 | 0.43 / 0.41 | 0.89 (0.84-0.95) | $2.29 \times 10^{-04}$ | 0.001 |

HLA, human leucocyte antigen; OR, odds ratio; 95%CI, confidence interval
[a] Obtained from multivariate unconditional logistic regression assuming an additive genetic model with sex and principal components as covariates
[b] Obtained from multivariate unconditional logistic regression assuming an additive genetic model with principal components as covariates
[c] For combined analyses we excluded 353 never-smoking women (85 cases and 268 controls) from Oncoarray's studies (Seoul and NJLCS) to avoid a possible overlap of samples.
[d] amino acids in tigh LD ($r^2 = 0.99$) with Glu-98 and Gln-10, part of HLA-DRB1*0701 and HLA-DRB1*0401 protein sequence and with the rs2395185 (reported by Lan et al.)
The study-wide significant threshold was $P = 6 \times 10^{-6}$ (Bonferroni correction)

markers. If we identified other independently associated markers, we included them also as covariates in subsequent conditional analyses. All statistical analyses were performed in R version 3.2.3 (https://www.r-project.org/).

To help the interpretation of the results and the comparison between ethnicities, supplementary tables 12 and 13 display statistical power calculations for both populations but also imputation probabilities for the associated variants.

**Meta-analysis of Asian datasets**. We combined the results of Asian Oncoarray and Lan et al. studies using GWAMA (Genome-Wide Association Meta-Analysis)[39] software to perform random-effect meta-analyses. The software incorporates error trapping, which facilities to identify strand alignment errors and allele flipping, and performs tests of heterogeneity of effects between studies.

**Haplotype inference, visualization and association analyses**. The results of HLA alleles and SNPs might not translate directly to a single locus as a result of the extended linkage disequilibrium (LD) known to exist in the MHC; it is possible that markers that seem to be acting independently with respect to genotype risk could be on a shared haplotype (http://www.ebi.ac.uk/ipd/imgt/hla/). We search for significant combination of HLA alleles using Haplo.stats package v.1.7.7 (https://cran.r-project.org/web/packages/haplo.stats/index.html) implemented in R software to generate population-based haplotypes from phased genotypes obtained from the imputed data. Haplo.stats uses the expectation-maximization algorithm and progressively inserts batches of loci into haplotypes. We inferred haplotypes frequencies within class I alleles, class II and along the MHC (class I and class II) for Asians and Europeans. Then, using the statistical framework and covariates defined above, we individually tested each of the haplotypes for association with lung cancer overall and in the subgroups described.

HLA linkage disequilibrium (LD) maps were obtained using Disentangler (http://kumasakanatsuhiko.jp/projects/disentangler/), a graphical tool suitable for visualization of haplotype configurations across multiallelic genetic markers for which typical triangular heat maps with LD indices will not work. Disentangler also applies expectation-maximization algorithms to estimate the haplotype frequencies between adjacent markers, and it uses then this information to determine the order of the alleles for each marker and the number of crossing lines between adjacent markers[40,41]. We created separate population-specific maps for cases and controls, using the whole collection for Asians and a randomized set of 5,000 cases and 5,000 controls for Europeans.

**Unravelling candidate functional variants**. Using the independently associated HLA alleles and/or haplotypes, their significant amino acids and SNPs, we searched for the best overall HLA model. First, we used a forward selection stepwise regression of significant amino acids within the independently associated regions. For each gene or region we looked for the set of significant amino acid positions and we defined the classical alleles with consistent residues at those positions. We also included as covariates in these risk models the independent markers previously identified. The aim was to find the best set of amino acids and/or HLA alleles that were independently associated with lung cancer as judged by the lowest AIC (Akaike Information Criterion) and BIC (Bayesian Information Criterion)[42,43].

Ribbon representations of the associated HLA molecules were constructed with the UCSF Chimera software[44] to locate significant amino acids in the tri-dimensional structure of the protein (http://www.cgl.ucsf.edu/chimera/). We also

used the online tool HaploReg v4.1[38] to confirm location of each SNP in relation to annotated protein-coding genes and non-coding regulatory elements (http://archive.broadinstitute.org/mammals/haploreg/haploreg.php).

**Data availability**

Genotype data for the lung cancer OncoArray study have been deposited at the database of Genotypes and Phenotypes (dbGaP) under accession phs001273.v1.p1. The Asian replication dataset was downloaded from dbGaP under accession phs000716.v1.p1.

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

## Acknowledgements

Transdisciplinary Research for Cancer in Lung (TRICL) research team of the International Lung Cancer Consortium (ILCCO) was supported by (U19-CA148127 and CA148127S1). The ILCCO data harmonization is supported by Cancer Care Ontario Research Chair of Population Studies to R. H. and Lunenfeld-Tanenbaum Research Institute, Sinai Health System. TRICL-ILCCO Oncoarray was supported by in-kind genotyping Centre for Inherited Disease Research (26820120008i-0-26800068-1). CAPUA study was supported by FIS-FEDER/Spain grant numbers FIS-01/310, FIS-PI03-0365, and FIS-07-BI060604, FICYT/Asturias grant numbers FICYT PB02-67 and FICYT IB09-133, and the University Institute of Oncology (IUOPA), of the University of Oviedo and the Ciber de Epidemiología y Salud Pública. CIBERESP, SPAIN. The work performed in the CARET study was supported by the The National Institute of Health/ National Cancer Institute: UM1 CA167462 (PI: Goodman), National Institute of Health UO1-CA6367307 (PIs Omen, Goodman); National Institute of Health R01 CA111703 (PI Chen), National Institute of Health 5R01 CA15198901A1(PI Doherty). Norway study was supported by Norwegian Cancer Society, Norwegian Research Council The NELCS study: Grant Number P20RR018787 from the National Center for Research Resources (NCRR), a component of the National Institutes of Health (NIH). The Liverpool Lung project is supported by the Roy Castle Lung Cancer Foundation. The Harvard Lung Cancer Study was supported by the NIH (National Cancer Institute) grants CA092824, CA090578, CA074386. The Multiethnic Cohort Study was partially supported by NIH Grants CA164973, CA033619, CA63464 and CA148127. The work performed in MSH-PMH study was supported by The Canadian Cancer Society Research Institute (020214), Ontario Institute of Cancer and Cancer Care Ontario Chair Award to RJH and GL and the Alan Brown Chair and Lusi Wong Programs at the Princess Margaret Hospital Foundation. NJLCS work was funded by the State Key Program of National Natural Science of China (81230067), the National Key Basic Research Program Grant (2011CB503805), the Major Program of the National Natural Science Foundation of China (81390543). Tampa Lung Cancer Study (Tampa) was supported by National Institutes of Health: R01 DE013158, PO1 CA68384 and R01 ES025460. The Shanghai Cohort Study (SCS) was supported by National Institutes of Health R01 CA144034 (PI: Yuan) and UM1 CA182876 (PI: Yuan). The Singapore Chinese Health Study (SCHS) was supported by National Institutes of Health R01 CA144034 (PI: Yuan) and UM1 CA182876 (PI: Yuan). TCL work has been supported in part the James & Esther King Biomedical Research Program (09KN-15), National Institutes of Health Specialized Programs of Research Excellence (SPORE) Grant (P50 CA119997), and by a Cancer Center Support Grant (CCSG) at the H. Lee Moffitt Cancer Center and Research Institute, an NCI designated Comprehensive Cancer Center (grant number P30-CA76292). The Vanderbilt Lung Cancer Study – BioVU dataset used for the analyses described was obtained from Vanderbilt University Medical Center's BioVU, which is supported by institutional funding, the 1S10RR025141-01 instrumentation award, and by the Vanderbilt CTSA grant UL1TR000445 from NCATS/NIH. Dr. Aldrich was supported by NIH/National Cancer Institute K07CA172294 (PI: Aldrich) and Dr. Bush was supported by NHGRI/NIH U01HG004798 (PI: Crawford). The L2 study. Lung cancer cases and controls were recruited through a multicentric case-control study coordinated by the International Agency for Research on Cancer in Russia, Poland, Serbia, Czech Republic, and Romania from 2005 to 2013. Cases were incident cancer patients collected from general hospitals. Controls were recruited from individuals visiting general hospitals and out-patient clinics for disorders unrelated to lung cancer and/or its associated risk factors, or from the general population. Information on lifestyle risk factors, medical and family history was collected from subjects by interview using a standard questionnaire. All study participants provided written informed consent. The current study included 1,133 lung cancer cases and 1,117 controls genotyped on the Oncoarray. The study in Lodz center was partially funded by Nofer Institute of Occupational Medicine, under task NIOM 10.13: Predictors of mortality from non-small cell lung cancer - field study.

## Author contributions

A.F.-I. contributed to data harmonization, HLA imputation, HLA quality control, formal analysis, writing original draft. C.L. contributed to data harmonization and HLA imputation. X.X., Y.L., D.C.Q., J.B., Y.H., A.K., D.Z., X.J. contributed to genotype calling and primary quality control filters. J.M., D.A., S.L., M.-S.T., A.T., A.F.-S., G.F.-T., C.C., J. D., G.G., S.E.B., M.T.L., M.J., J.K.F., M.D., H.B., H.-E.W., A.R., D.C., G.R., S.A., S.S., X.W., O.M., H.B., L.L.M., X.Z., G.L., A.A., E.D., L.A.K., E.H.F.M.H., H.S., J.D., A.H., S.Z., M.J., K.G., N.C., P.W., M.T., G.S., D.Z., A.M., M.K., S.O., J.L., M.S., B.S., V.J., I.H., C.B., M.S., M.O., Y.-C.H., J.-M.Y., P.L., M.B.S., M.C.A. contributed to data collection. R.J.H. coordinated the Lung Cancer OncoArray, is responsible for phenotype data harmonization across OncoArray studies and contributed to data collection. C.I.A. designed and coordinated the Lung Cancer OncoArray, and contributed to data collection. P.B. contributed to conceptualization, contributed to data collection, supervision, project

administration, funding acquisition, and provided overall supervision and management. All authors contributed to reviewing and editing original draft.

### Additional information

**Competing interests:** The authors declare no competing interests.

Aida Ferreiro-Iglesias[1], Corina Lesseur[1], James McKay[1], Rayjean J. Hung[2], Younghun Han[3], Xuchen Zong[2], David Christiani[4], Mattias Johansson[1], Xiangjun Xiao[3], Yafang Li[3], David C. Qian[3], Xuemei Ji[3], Geoffrey Liu[2], Neil Caporaso[5], Ghislaine Scelo[1], David Zaridze[6], Anush Mukeriya[6], Milica Kontic[7], Simona Ognjanovic[8], Jolanta Lissowska[9], Małgorzata Szołkowska[10], Beata Swiatkowska[11], Vladimir Janout[12], Ivana Holcatova[13], Ciprian Bolca[14], Milan Savic[15], Miodrag Ognjanovic[8], Stig Egil Bojesen[16,17,18], Xifeng Wu[19], Demetrios Albanes[5], Melinda C. Aldrich[20], Adonina Tardon[21], Ana Fernandez-Somoano[21], Guillermo Fernandez-Tardon[21], Loic Le Marchand[22], Gadi Rennert[23], Chu Chen[24], Jennifer Doherty[24,25], Gary Goodman[26], Heike Bickeböller[27], H-Erich Wichmann[28,29,30], Angela Risch[31,32,33], Albert Rosenberger[27], Hongbing Shen[34], Juncheng Dai[34], John K. Field[35], Michael Davies[35], Penella Woll[36], M. Dawn Teare[37], Lambertus A. Kiemeney[38], Erik H.F.M. van der Heijden[38], Jian-Min Yuan[39], Yun-Chul Hong[40], Aage Haugen[41], Shanbeh Zienolddiny[41], Stephen Lam[42], Ming-Sound Tsao[43], Mikael Johansson[44], Kjell Grankvist[45], Matthew B. Schabath[46], Angeline Andrew[3], Eric Duell[47], Olle Melander[48,49], Hans Brunnström[50], Philip Lazarus[51], Susanne Arnold[52], Stacey Slone[52], Jinyoung Byun[3], Ahsan Kamal[3], Dakai Zhu[3], Maria Teresa Landi[5], Christopher I. Amos[3] & Paul Brennan[1]

[1]International Agency for Research on Cancer, World Health Organization, Lyon 69372 cedex 08, France. [2]Lunenfeld-Tanenbaum Research Institute of Sinai Health System, University of Toronto, Toronto M5G 1X5, Canada. [3]Biomedical Data Science, Geisel School of Medicine at Dartmouth, Hanover 03755 NH, USA. [4]Department of Environmental Health, Harvard TH Chan School of Public Health, Massachusetts General Hospital/ Harvard Medical School, Boston 02115 MA, USA. [5]Division of Cancer Epidemiology and Genetics, National Cancer Institute, National Institutes of Health, Bethesda 20892-9768 MD, USA. [6]Russian N.N. Blokhin Cancer Research Centre, Moscow 115478, Russian Federation. [7]Clinical Center of Serbia, Belgrade 11000, Serbia. [8]International Organization for Cancer Prevention and Research, Belgrade 11070, Serbia. [9]M. Sklodowska-Curie Cancer Center, Institute of Oncology, Warsaw 02-034, Poland. [10]Department of Pathology, National Tuberculosis and Lung Diseases Research Institute, Warsaw 01-138, Poland. [11]Department of Environmental Epidemiology, Nofer Institute of Occupational Medicine, Lodz 91-348, Poland. [12]Faculty of Medicine, University of Olomouc, Olomouc 701 03, Czech Republic. [13]2nd Faculty of Medicine, Institute of Public Health and Preventive Medicine, Charles University, Prague CZ 128 00, Czech Republic. [14]Institute of Pneumology "Marius Nasta", Bucharest RO-050159, Romania. [15]Department of Thoracic Surgery Clinical Center of Serbia Belgrade, Belgrade 11000, Serbia. [16]Copenhagen General Population Study, Herlev and Gentofte Hospital, Copenhagen 2730, Denmark. [17]Department of Clinical Biochemistry, Herlev and Gentofte Hospital, Copenhagen University Hospital, Copenhagen 2730, Denmark. [18]Faculty of Health and Medical Sciences, University of Copenhagen, Copenhagen 2730, Denmark. [19]Department of Epidemiology, The University of Texas MD Anderson Cancer Center, Houston 77030 TX, USA. [20]Department of Thoracic Surgery, Division of Epidemiology, Vanderbilt University Medical Center, Nashville 37232-4682 TA, USA. [21]University of Oviedo and CIBERESP, Faculty of Medicine, Oviedo 33006, Spain. [22]Epidemiology Program, University of Hawaii Cancer Center, Honolulu 96813 HI, USA. [23]Clalit National Cancer Control Center at Carmel Medical Center and Technion Faculty of Medicine, Haifa 3525433, Israel. [24]Department of Epidemiology, University of Washington School of Public Health and Community Medicine, Seattle 98195 WA, USA. [25]Fred Hutchinson Cancer Research Center, Seattle 98109 WA, USA. [26]Swedish Medical Group, Seattle 98104 WA, USA. [27]Department of Genetic Epidemiology, University Medical Center, Georg-August-University Göttingen, Göttingen 37073, Germany. [28]Institute of Medical Informatics, Biometry and Epidemiology, Chair of Epidemiology, Ludwig Maximilians University, Munich D-85764, Germany. [29]Helmholtz Center Munich, Institute of Epidemiology 2, Munich D-85764, Germany. [30]Institute of Medical Statistics and Epidemiology, Technical University Munich, Munich D-80333, Germany. [31]University of Salzburg and Cancer Cluster Salzburg, Salzburg 5020, Austria. [32]Translational Lung Research Center Heidelberg (TLRC-H), Heidelberg 69120, Germany. [33]German Center for Lung Research (DZL), Heidelberg 69121, Germany. [34]Department of Epidemiology and Biostatistics, Jiangsu Collaborative Innovation Center for Cancer Medicine, School of Public Health, Nanjing Medical University, Nanjing 211166, China. [35]Institute of Translational Medicine, University of Liverpool, Liverpool L3 9TA, UK. [36]Department of Oncology, University of Sheffield, Sheffield S10 2RX, UK. [37]School of Health and Related Research, University Of Sheffield, England S1 4DA, UK. [38]Radboud University Medical Center,

Nijmegen 9500, The Netherlands. [39]University of Pittsburgh Cancer Institute, Pittsburgh 15232 PA, USA. [40]Department of Preventive Medicine, Seoul National University College of Medicine, Seoul 110-799, Republic of Korea. [41]National Institute of Occupational Health, Oslo N-0033, Norway. [42]British Columbia Cancer Agency, Vancouver V5Z 1M9, Canada. [43]Princess Margaret Cancer Centre, Toronto ON M5G 1L7, Canada. [44]Department of Radiation Sciences, Umeå University, Umeå 901 85, Sweden. [45]Department of Medical Biosciences, Umeå University, Umeå 901 85, Sweden. [46]Department of Cancer Epidemiology, H. Lee Moffitt Cancer Center and Research Institute, Tampa 33612 FL, USA. [47]Unit of Nutrition and Cancer, Catalan Institute of Oncology (ICO-IDIBELL), Barcelona 08908, Spain. [48]Department of Clinical Sciences Malmö, Lund University, Malmö 221 00, Sweden. [49]Department of Internal Medicine, Skåne University Hospital, Malmö, Sweden. [50]Laboratory Medicine Region Skåne, Department of Clinical Sciences Lund, Pathology, Lund University, Lund 221 00, Sweden. [51]Department of Pharmaceutical Sciences, College of Pharmacy, Washington State University, Spokane 99202 WA, USA. [52]University of Kentucky, Markey Cancer Center, Lexington 40536-0098 KY, USA

