## [Peer Review File · Nature Communications]

Reviewers' comments:

Reviewer #1 (Remarks to the Author):

Authors have done a fine mapping analysis of HLA variants in lung cancer through imputation-based association analysis. The analysis was done by using a very large European sample of 18,924 cases and 15,439 controls, but a much smaller Asian sample of 2324 cases and 1656 controls. The analysis identified two independent associations (HLA-B*0801/AH8.1 and HLA-DQB1*06 in the European sample and three independent associations (HLA-DQB1*0401, DRB1*0701 and an intronic SNP rs2256919 in HLA-A). Authors also compared the results from the European and Asian samples and failed to identify any overlapping loci between the two populations. The study was well designed, and the finding has advanced our understanding on the lung cancer association within the MHC region, at least in European population. There are several issues/concerns that need to be addressed before considering for publication:

1. P value of $0.05/8,291 = 6 \times 10^{-6}$ was used as the threshold for study-wide significance after considering the highest number of HLA variants that was included in the analysis. However, it is clear that both HLA variants and other SNPs within the MHC region were included in analysis. So, the correction should be done for the number of both HLA variants and SNPs. In addition, the association was done in European and Asian samples separately. So, the P value should also be corrected for two independent analyses of different ethnic samples.
2. The results from the Asian sample are not convincing, giving that it was done in a very small sample, and the evidence were just barely below the threshold of significance, which is particularly concerning given that the current threshold of significance is not stringent enough (see above). Using a more stringent threshold, the evidences from the Asian sample may not be statistically significant. In addition, the imputation of the Asian sample was done by using a very small reference panel, and the quality of imputation was much lower than the analysis of European samples. The imputation analysis of the Asian sample should be done by including the large Chinese HLA reference panel published in Nature Genetics last year.
3. More information need to be provided regarding to the effort on controlling population stratification. The control was done by including PCs, but not information was provided on the number of PCs that were included in the analysis. The genome-wide analysis of independent SNPs should have been performed to evaluate the effectiveness of population stratification controlling in both European and Asian samples.
4. Given that a large number of independent samples were included in the analysis, genetic heterogeneity should have been evaluated.
5. Due to the concern on the sample size and thus the power of Asian sample, the conclusion of no overlapping between Asian and European hits is premature. In particular, it is premature to claim that the current findings may suggest that different antigens in the two populations may have led to the involvement of different HLA alleles in two populations.
6. Authors claimed that smoking status has no effect on the associations identified in this study. However, the results in the Supplementary Figure 1 are not totally consistent with the claim. For example, AH8.1 showed risk effect in the overall analysis as well as the smoking groups, but protective effect (OR = 0.88) in the group of never smoking group. Although the result from the group of never smoking is not statistically significant, the opposite OR values are concerning. It will be important to report the association results of all the reported variants after including smoking as co-variate in the analysis.

7. It will be interesting to do a stratified analysis using EGFR status, if the information is available.

Reviewer #2 (Remarks to the Author):

Overall impression:

The authors tested SNP association for MHC molecules across cases and controls from large cohorts. The group analyzed genetic variation among 21k lung cancer patients and 17k controls using the Illumina OncoArray. They imputed sequence variation in classical HLA genes, and mapped the MHC association for lung cancer risk and major histologies and compared results between ethnicities. The findings are summarized clearly and are clearly novel. Several important HLA genotypes were discovered, and the findings have implications for future lung cancer immunotherapies targeting the HLA antigen. The results implicate several HLA–tumor peptide interactions as the major MHC factor modulating lung cancer susceptibility. This manuscript attempts to understand the role of MHC in risk assessment for lung cancer. In the context of the success of immunotherapy, this brings relevance to the field and is very timely and the results move the field forward. However, this is a descriptive report. There is no biology experiment validation or supporting evidence from other papers to confirm the relevance of the findings.

Major concerns:

The results suggest that the genetic risk of the MHC region on lung cancer is different between Asians and Europeans. The contribution of this HLA based genetic risk to lung cancer development remains to be determined and expected to be low.

In this article, one of the key findings is that the association is different between Asian and European populations (European population to SCC and Asian population to AD). But there are a lot of differences between European and Asian population in this study. The number of samples in two populations were significantly different, which may result in different statistical power of association detection. The smoking status, the histology groups were also different. So it is very hard to conclude the difference is because of ethnic-specific genetic effects.

The statistical analysis was the most important part of this article, but it is difficult to follow. Which variables (besides Sex) were included? As mentioned in the method, "We assessed the association between the described variants and lung cancer risk, as well as predominant histological types and smoking behavior." Results related to association between variants and histological types or smoking status are missing.

The presence of a threonine in the amino acid position 163 of HLA-B*0801 accounts for the main part of this effect as it fitted the data in the conditional model. The biological significance remains elusive. Limitations of this paper in regards to the functional significance of these findings need to be further discussed. The associations between HLA alleles and risk for lung cancer have not been validated functionally. The OR associated with risk remain small. The value of these SNPs has not been tested in a prediction model for lung cancer risk which is one of the translational goals of this research.

Figures 1&2 need some work. The color code is difficult to read the figure is low resolution. The light and deep blue colors are difficult to separate. The green color does not show. I can't find green dots at the left panel, but there are many black dots which were not in the legend. In Figure 2, the HLA genes should be labeled under the red and green dots, but they were not labeled in the right position.

The number of sample size should be provided in each figure and table, including conditional logistic regression in Figure 1 and 2, Table 1, 2, 3, 4. These were very important for interpreting the results and tables.

More details should be provided in the "Supplement Method" section to help understand the

experiment methods and reproduce the results. For example: a. how many probes and samples were kept for further analysis after QC and imputation? b. The statistical analysis was performed in which software? c. When mentioning software (Disentangler, Haplo.stats), the version number should be provided as they were updated very frequently and the results were related to software version; are the p values used in all figures and tables adjusted p values?

Minor concerns:

1. The significant threshold was set as $P = 6 \times 10^{-6}$ in two populations. Any consideration for setting different significant threshold for two populations? As the p values were influenced by sample size.
2. Page 8, Line 203, there was no "Supplementary Figure 5a".
3. Page 10 Line 260, "0,76" should be "0.76";
4. In Supplement Figure 2, I can't find "AH8.1 (yellow) is clearly shown, whose frequency is increased in cases regarding controls". I will suggest adding the label for Y axis (Frequency) to make it more clear.
5. In Table 4, there were no "AIC" results, only "BIC" results were included.
6. The analysis was based on NCBI build 36, which was released more than 10 years ago. The human genome has changed a lot. Any consideration using a newer version of genome?

Reviewer #3 (Remarks to the Author):

This paper addresses an important subject in mapping susceptibility alleles for lung cancer in large meta-analyses- the role of HLA in lung cancer susceptibility. The results are preliminary and quite interesting but do not fully support the title. The study is predominately focused on European ancestry and African Americans is barely touched while the Asian sample size is quite small and in a non-comparable group (non-smokers). In this regard, this reviewer finds the title misleading. There are a number of major issues that should be addressed in a subsequent revised version.

1. The authors should carefully temper the contention that HLA imputation is stable and accurate. Many would disagree with this contention- which is slightly but inadequately discussed in the paper. The complexity of HLA is such that many regions are hard to impute- especially because of gene conversion.
2. Pursuant to #1, to suggest the findings based on imputed data and not conduct critical and necessary validation studies is highly problematic. For basic GWAS, nearly all conduct validation studies and for HLA, it is very important to do this- whether it is sequencing or SNP tests. The bottom line is that this is required despite the contentions of those who published their method.
3. Smoking a major risk factor for lung cancer in Europeans and less so for the available Asian nonsmoking women (with distinct regions) but the analyses are not thorough and fully characterized. More details on the types of analyses as well as the effect of smoking is required. It is hard to know where it is an adjustment versus a confounder versus actually studied independently. There are several studies that suggest the genetic contribution to smoking phenotypes differs between European and Asian ancestry- yet this is not addressed in discussion or in the analyses.
4. The extensive discussion of HLA grooves and structure is interesting and predicated on one tool- again one that others might not agree as the optimal. This should be discussed. Shortening this discussion might be an enhancement of the manuscript as the interpretation is quite long when no new data is presented.
5. It is hard to say that a fair comparison across ethnic groups has been conducted so the title is not well supported. This reviewer strongly encourages the focus to go deeper into the European data and make this the central focus. The numbers of Asians is smaller and less compelling- small enough that it is hard to make much of a set of conclusions.

Reviewer #1 (Remarks to the Author):

Authors have done a fine mapping analysis of HLA variants in lung cancer through imputation-based association analysis. The analysis was done by using a very large European sample of 18,924 cases and 15,439 controls, but a much smaller Asian sample of 2324 cases and 1656 controls. The analysis identified two independent associations (HLA-B*0801/AH8.1 and HLA-DQB1*06 in the European sample and three independent associations (HLA-DQB1*0401, DRB1*0701 and an intronic SNP rs2256919 in HLA-A). Authors also compared the results from the European and Asian samples and failed to identify any overlapping loci between the two populations. The study was well designed, and the finding has advanced our understanding on the Lung cancer association within the MHC region, at least in European population. There are several issues/concerns that need to be addressed before considering for publication:

1. P value of $0.05/8,291 = 6 \times 10^{-6}$ was used as the threshold for study-wide significance after considering the highest number of HLA variants that was included in the analysis. However, it is clear that both HLA variants and other SNPs within the MHC region were included in analysis. So, the correction should be done for the number of both HLA variants and SNPs. In addition, the association was done in European and Asian samples separately. So, the P value should also be corrected for two independent analyses of different ethnic samples.

We thank the reviewer for this comment. We actually used both HLA variants and SNPs within the MHC region to calculate the threshold for study significance. These account for a total of 8,291 variants in the European series, obtained by the sum of the genotyped SNP variants used for the imputation, as well as the imputed SNPs, HLA alleles and amino acid variants after passing quality control. Therefore, we maintained the p-value cut-off for Europeans in 6×10^{-6} . The final number of markers analysed for Asian analysis after quality control including genotyped and imputed variants was 5,404. Following the same rationale we applied in Europeans, a p-value cut off of $0.05/5,404 = 9 \times 10^{-6}$ should be applied, but we decided to maintain the more stringent 6×10^{-6} threshold for study-wide significance in Asians, in both the samples from Oncoarray and the ones from the replication set that we included. These Bonferroni corrections imply that all variants are independent, whereas clearly they are not. The corrected p-values are therefore already very conservative.

We added a note clarifying this point in Methods section (line 75-80, page 3).

2. The results from the Asian sample are not convincing, giving that it was done in a very small sample, and the evidence were just barely below the threshold of significance, which is particularly concerning given that the current threshold of significance is not stringent enough (see above). Using a more stringent threshold, the evidences from the Asian sample may not be statistically significant. In addition, the imputation of the Asian sample was done by using a very small reference panel, and the quality of imputation was much lower than the analysis of European samples. The imputation analysis of the Asian sample should be done by including the large Chinese HLA reference panel published in Nature Genetics last year.

Regarding the threshold used, and as explained above, this correction is already very conservative and far exceeds the number of independent tests expected within this region given its high linkage disequilibrium (methods; line 75-80, page 3)

We understand the reviewer's concern about the Asian sample size, and the resulting p-values being close to the threshold. In order to increase the sample size of the Asian analysis we included an additional set of 8,537 samples obtained from a published GWA study of lung cancer in Asia (Lan et al, Nature Genetics 2012). This is a multicentre dataset of non-smoking women from mainland China, South Korea, Japan, Singapore, Taiwan and Hong Kong (Supplementary Figure 1, Supplementary Tables 8 and 9). We conducted a meta-analysis of both Asian datasets and then extended our comparison between the European and Asian ethnicity studies. Results from replication and combined analyses are displayed in Table 5 and Supplementary Figure 4. We also modified Methods, Results and Discussion sections as appropriate. Our observations suggest that, after meta-analysis with the new Asian results, the lack of overlap between Asian and European hits becomes even clearer. In the Asian replication collection, we also observed associations between HLA class II alleles and risk of lung adenocarcinoma only. We observed two of the three independent HLA allele effects that were initially observed. One is represented by the four-digit allele *HLA-DQB1*0401* as in the discovery, and the other is better explained by the association of Ala-104/Glu-98/Gln-10, part of *HLA-DRB1*0701* and *HLA-DRB1*0401* protein sequence. This is the same effect as the one described by Lan et al. in the intergenic region in 6p21.32 (rs2395185) as all SNP and amino acids changes are in tight LD ($r^2=0.99$). In summary, our updated findings maintain the same conclusion pointing to a different role of HLA variants by population (Asian vs European) and by histology.

As the reviewer noted, a larger reference panel could improve the overall imputation quality. This would be apparent for rare variant imputation although less so for common variants that are already well imputed (75% of the common variants and 98% of common HLA alleles (MAF>0.05) had $R^2 \geq 0.9$). Given our sample size, our primary focus was on common variants for which the current imputation is satisfactory. We did however try the reviewer's suggestion of imputing our data using the large reference panel proposed (Zhou et al., 2016). Unfortunately, due to technical issues related to the intensive computational requirements of the analysis using the large HLA HAN panel, the outcome was not satisfactory. Only 52,5% of the all variants passed overall imputation quality control using HAN-panel with a $R^2 \geq 0.7$. In particular, imputation quality for HLA alleles was very poor, with only 40% of common HLA alleles having an $R^2 \geq 0.7$ and only the 17% of all HLA alleles.

3. More information need to be provided regarding to the effort on controlling population stratification. The control was done by including PCs, but not information was provided on the number of PCs that were included in the analysis. The genome-wide analysis of independent SNPs should have been performed to evaluate the effectiveness of population stratification controlling in both European and Asian samples.

We included a more detailed description of population stratification correction in the Methods (line 21-26, page 1). Briefly, PCA was done in the context of a recent genome-wide analysis using the OncoArray platform (McKay et al. Nature genetics 2017). Population stratification analysis (implemented in FlashPCA software) identified 10 and 3 significant eigenvectors on the European

and the Asian dataset, respectively. We have referenced now the original study (line 13-15, page 1) and added the plots showing the top principal components for Europeans and Asians (Supplementary Figure 5).

Details for the PCA of the new Asian dataset are also included in the Methods (21-36, page 1-2) and plots in Supplementary Figure 5.

4. Giving that a large number of independent samples were included in the analysis, genetic heterogeneity should have been evaluated.

Indeed, 30 separate epidemiological studies were included, although often with limited information on ethnic origin. For this reason, and as is standard practice in large population based genetic studies, we have used eigenvectors in lieu of study of recruitment to better adjust for genetic/ethnic origin. Generalized linear models show that the eigenvectors were significantly associated with the study of recruitment either in both Asian and European series (p-value: $< 2.2 \times 10^{-16}$). This was also clarified in the Methods section (line 24-26, page 1).

5. Due to the concern on the sample size and thus the power of Asian sample, the conclusion of no overlapping between Asian and European hits is premature. In particular, it is premature to claim that the current findings may suggest that different antigens in the two populations may have led to the involvement of different HLA alleles in two populations.

It is well-know that population history and different exposures shape the specific HLA profile of each population. As discussed in the comment #2, is not surprising that our updated findings maintain the same conclusion pointing to a possible different role of HLA variants by population and by histology. However, results for the Asian squamous-cell carcinoma data are limited by the sample size (line 377, page 12). Even we doubled the number of the initial samples we still cannot rule out that the observed differences for this histology subtype could be due to the lack of power in the Asian collection.

6. Authors claimed that smoking status has no effect on the associations identified in this study. However, the results in the Supplementary Figure 1 are not totally consistent with the claim. For example, AH8.1 showed risk effect in the overall analysis as well as the smoking groups, but protective effect (OR = 0.88) in the group of never smoking group. Although the result from the group of never smoking is not statistic significant, the opposite OR values are concerning. It will be important to report the association results of all the reported variants after including smoking as co-variate in the analysis.

We agree with the reviewer on checking further the possible influence of tobacco on AH8.1 association and the rest of the variants. We performed association analyses including smoking as a co-variate for all the associated markers. The Supplementary tables 8 and 9 shows these results for Europeans and Asians, respectively. As can be seen, the results are extremely similar to the original results in Tables 2 and 3, indicating that adjustment by smoking makes little difference. There are some minor fluctuations in the p-values and ORs, as expected. We included a comment on this analysis (line 237-245, page 7).

Regarding genetic heterogeneity in the stratified analyses, we also added heterogeneity *P*-values calculated using Cochran's Q test to Supplementary figure 1, as it was not included in the original version. We have also commented specifically on the heterogeneity observed for AH8.1 by smoking status (line 239, page 7).

7. It will be interesting to do a stratified analysis using EGFR status, if the information is available.

We thank the reviewer for this suggestion. We agree that it will add interesting information but unfortunately EGFR status is not available for this data.

Reviewer #2 (Remarks to the Author):

Overall impression:

The authors tested SNP association for MHC molecules across cases and controls from large cohorts. The group analyzed genetic variation among 21k lung cancer patients and 17k controls using the Illumina OncoArray. They imputed sequence variation in classical HLA genes, and mapped the MHC association for lung cancer risk and major histologies and compared results between ethnicities. The findings are summarized clearly and are clearly novel. Several important HLA genotypes were discovered, and the findings have implications for future lung cancer immunotherapies targeting the HLA antigen. The results implicate several HLA–tumor peptide interactions as the major MHC factor modulating lung cancer susceptibility. This manuscripts attempts to understand the role of MHC in risk assessment for lung cancer. In the context of the success of immunotherapy, this brings relevance to the field and is very timely and the results move the field forward. However, this is a descriptive report.

There is no biology experiment validation or supporting evidence from other papers to confirm the relevance of the findings.

Major concerns:

The results suggest that the genetic risk of the MHC region on lung cancer is different between Asians and Europeans. The contribution of this HLA based genetic risk to lung cancer development remains to be determined and expected to be low.

In this article, one of the key finding is that the association is different between Asian and European populations (European population to SCC and Asian population to AD). But there are a lot of difference between European and Asian population in this study. The number of samples in two populations were significantly different, which may result in different statistical power of association detection. The smoking status, the histology groups were also different. So it is very hard to conclude the difference is because of ethnic-specific genetic effects.

We understand the reviewer's concern about the Asian sample size, and the resulting p-values being close to the threshold. In order to increase the sample size of the Asian analysis we included an additional set of 8,537 samples obtained from another published GWA study of lung cancer in Asia (Lan et al, Nature Genetics 2012). This is a multicentre dataset of non-smoking women from mainland China, South Korea, Japan, Singapore, Taiwan and Hong Kong (Supplementary Figure 1, Supplementary Tables 8 and 9). We conducted a meta-analysis of both Asian datasets and then extended our comparison between the European and Asian ethnicity studies. Results from replication and combined analyses are displayed in Table 5 and Supplementary Figure 4. We also modified Methods, Results and Discussion sections as appropriate. Our observations suggest that, after meta-analysis with the new Asian results, the lack of overlap between Asians and European hits becomes even clearer. In the Asian replication collection, we also observed associations between HLA class II alleles and risk of lung adenocarcinoma only. We observed two of the three independent HLA allele effects that were initially present. One is represented by the four-digit allele *HLA-DQB1*0401* as in the discovery, and the other is better explained by the association of Ala-104/Glu-

98/Gln-10, part of *HLA-DRB1*0701* and *HLA-DRB1*0401* protein sequence. This is the same effect as the one described by Lan et al. in the intergenic region in 6p21.32 (rs2395185) as all SNP and amino acids changes are in tight LD ($r^2=0.99$). In summary, our updated findings maintain the same conclusion pointing to a different role of HLA variants by population (Asian vs European) and by histology.

On top of this, It is well-know that population history and different exposures shape the specific HLA profile of each population. It's not surprising that our updated findings maintain the same conclusion pointing to a possible different role of HLA variants by population and by histology. However, results for the Asian squamous-cell carcinoma data are limited by the sample size (line 377, page 12). Even we doubled the number of the initial samples we still cannot rule out that the observed differences for this histology subtype could be due to the lack of power in the Asian collection.

The statistical analysis was most important part of this article, but it is difficult to follow. Which variables (besides Sex) were included? As mentioned in the method, "We assessed the association between the described variants and lung cancer risk, as well as predominant histological types and smoking behavior." Results related to association between variants and histological types or smoking status are missing.

We included a more detailed explanation of the analyses performed (Methods, line 21-26 and 31-36 and 64, page 1-2). Significant results from stratified analyses are shown in supplementary Figure 1 and referenced in the main text (line 185-195; page 7).

The presence of a threonine in the amino acid position 163 of HLA-B*0801 accounts for the main part of this effect as it fitted the data in the conditional model. The biological significance remains elusive. Limitations of this papers in regards to the functional significance of these findings need to be further discussed. The associations between HLA alleles and risk for lung cancer have not been validated functionally. The OR associated with risk remain small. The value of these SNPs has not been tested in a prediction model for lung cancer risk which is one of the translational goals of this research.

We noted that the associated OR described are small, but on the other hand this is as expected for a complex phenotype such as lung cancer. While a functional validation would be ideal, we did replicate the Asian results in an independent collection. Although this would be desirable for Europeans as well, our recruitment strategy was extensive, involving contacts with all major groups that have collected large lung cancer case-control studies with blood samples, and identifying further larger studies of these cancer types is not currently feasible.

We do recognize the reviewers point, so have stated the need for future functional analyses of our findings in the main text (line 236, page 11).

Figures 1&2 need some work. The color code is difficult to read the figure is low resolution. The light and deep blue colors are difficult to separate. The green color does not show. I can't find green dots at the left panel, but there are many black dots which were not in the legend. In Figure

2, the HLA genes should be labeled under the red and green dots, but they were not labeled in the right position.

All figures were redone and legend was corrected too. Black dots correspond to variants not mapped. Note that HLA alleles (green dots) are much less than other markers so is easy that in some figures due to the dense number of variants and the LD they can be hidden behind others markers

The number of sample size should be provided in each figure and table, including conditional logistic regression in Figure 1 and 2, Table 1, 2, 3, 4. These were very important for interpreting the results and tables.

We included sample sizes in each of the tables referred to by the reviewer. Figure 1 and 2 already included sample size in the upper-right part of each figure on the left but we made it bigger to help the reader; sample size of conditional logistic regression plots, on the right of each figure, correspond to the same number of highlighted plot on the left as is representing the same logistic regression but including the specified markers as covariates.

More details should be provided in the “Supplement Method” section to help understand the experiment methods and reproduce the results. For example:

We have updated Methods sections according all the reviewer suggestions.

a. how many probes and samples were kept for further analysis after QC and imputation?

A total of 8,291 variants were used for the analysis after QC ($R^2 > 0.7$) for Europeans. For Asians 5,504 variants were kept after QC ($R^2 > 0.7$) (Methods; line 75-81, page 3)

b. The statistical analysis was performed in which software?

Statistical analyses were performed in R version 3.2.3 (Methods, line 86, page 3) and (main text, line 406, page 14)

c. When mentioning software (Disentangler, Haplo.stats), the version number should be provided as they were updated very frequently and the results were related to software version; are the p values used in all figures and tables adjusted p values?

We included the technical details the reviewer asked for as it was possible:

-Disentangler has one release, thus this software will stay referenced as it was done in the original version.

-haplo.stats package version was included in the text (Methods, line 98, page 4).

-p values used are not adjusted for multiple testing. We included this information and the cut-off of the study in the footnote of each figure and table.

Minor concerns:

1. The significant threshold was set as $P= 6 \times 10^{-6}$ in two populations. Any consideration for setting different significant threshold for two populations? As the p values were influenced by sample size.

We actually used both HLA variants and SNPs within the MHC region to calculate the threshold for study significance. These accounts for a total of 8,291 variants in the European series, obtained by the sum of the genotyped SNPs variants used for the imputation, as well as the imputed SNPs, HLA alleles and amino acid variants after passing quality control. Therefore, we maintained the p-value cut-off for Europeans in 6×10^{-6} . The final number of markers analysed for Asian analysis after quality control including genotyped and imputed variants was 5,404. Following the same rationale we applied in Europeans, a p-value cut off of $0.05/5,404 = 9 \times 10^{-6}$ should be applied, but we decided to maintain the more stringent 6×10^{-6} threshold for study-wide significance in Asians, in both the samples from Oncoarray and the replication set that we included. These Bonferroni corrections imply that all variants are independent, whereas clearly they are not. The corrected p-values are therefore already very conservative. (Methods; line 75-81, page 3)

2. Page 8, Line 203, there was no "Supplementary Figure 5a".

We have corrected the typo. The reference was Supplementary Figure 1a.

3. Page 10 Line 260, "0,76" should be "0.76";

We have corrected the typo accordingly.

4. In Supplement Figure 2, I can't find "AH8.1 (yellow) is clearly shown, whose frequency is increased in cases regarding controls". I will suggest adding the label for Y axis (Frequency) to make it more clear.

We have labelled Y axis accordingly.

5. In Table 4, there were no "AIC" results, only "BIC" results were included.

As we argued in the main text (line 274-276, page 8-9) we prefer to show just the results from BIC criterion in the main tables. We deleted the reference to AIC criterion in Table 4 to avoid confusions.

6. The analysis was based on NCBI build 36, which was released more than 10 years ago. The human genome has changed a lot. Any consideration using a newer version of genome?

Actually, Oncoarray is in NCBI Build 37. Build 36 genome coordinates were obtained after imputation from the imputation panel.

Reviewer #3 (Remarks to the Author):

This paper addresses an important subject in mapping susceptibility alleles for lung cancer in large meta-analyses- the role of HLA in lung cancer susceptibility. The results are preliminary and quite interesting but do not fully support the title. The study is predominately focused on European ancestry and African Americans is barely touched while the Asian sample size is quite small and in a non-comparable group (non-smokers). In this regard, this reviewer finds the title misleading. There are a number of major issues that should be addressed in a subsequent revised version.

We thank the reviewer for this comment. We agree on changing the title as we did not do trans-ethnic analyses but just a comparison between European and Asian results. We would also point out the increased sample size for the Asian analysis in our resubmission.

1. The authors should carefully temper the contention that HLA imputation is stable and accurate. Many would disagree with this contention- which is slightly but inadequately discussed in the paper. The complexity of HLA is such that many regions are hard to impute- especially because of gene conversion.

The HLA imputation method used in this paper has been already validated (Jia et al., 2013) and widely used (Raychoudhuri et al. 2012, Lesueur et al. 2017) for the reference panels employed here and HLA genes analysed. The pre-training of the data and the statistical method is computationally intensive and ordinarily only needs to be performed once for the parameters mentioned if the data fits the ethnicity of the reference panel. We have therefore provided details of the astringent ethnic-filtering and evidence for how our samples clustered with the reference panel samples. Briefly, high quality data for the imputation process was obtained by applying standard quality control procedures where we excluded underperforming genotyping assays (judged by success rate, genotype distributions deviated from that expected by Hardy Weinberg equilibrium) and additionally, individuals with low genotyping success rate (<95%) and those with at least ≥90% of European or Asian ethnic origin judged by PCA using ancestry informative markers from OncoArray GWAs in lung cancer (McKay et al., 2017). Moreover, the Oncoarray chip gives a high SNP coverage in the HLA region (over 10,000 SNPs) which helps to perform a good quality HLA inference.

Gene conversion is indeed a problem to obtain good imputation in the HLA region at the individual level but not at population level. This method may not be reliable for donor matching, for example, but it is enough for population-based studies.

2. Pursuant to #1, to suggest the findings based on imputed data and not conduct critical and necessary validation studies is highly problematic. For basic GWAS, nearly all conduct validation studies and for HLA, it is very important to do this- whether it is sequencing or SNP tests. The bottom line is that this is required despite the contentions of those who published their method.

We understand the reviewer's concern but this is a secondary analysis coming from the largest and most extensive published GWA study of lung cancer (McKay et al. Nat Gen 2017). We did replicate the Asian results in an independent collection. Although this would be desirable for Europeans as well, the original lung cancer OncoArray study involved all major large case-control studies with blood samples (close to 30,000 lung cancer cases), and identifying further larger studies of these cancer types is simply not feasible. We do hope to increase the sample size of this lung cancer GWAS in the future, although it will take several years for ongoing recruitment efforts to identify the required numbers of cases.

We also re-imputed a random 10% subset of the samples using another pre-trained referenced panel and another statistical method called HIBAG (Zheng et al., 2014) which employs multiple expectation-maximization-based classifiers to estimate the likelihood of HLA alleles. The success of this method was similar and the correlation of the results after QC was of a $> 0, 95$. Additionally, our findings have also been observed in other cancers and autoimmune diseases which grant biological plausibility to our results. (methods; line 52-62, page 2, supplementary table 10 and main text; line 197-200, page 6)

3. Smoking a major risk factor for lung cancer in Europeans and less so for the available Asian nonsmoking women (with distinct regions) but the analyses are not thorough and fully characterized. More details on the types of analyses as well as the effect of smoking is required. It is hard to know where it is an adjustment versus a confounder versus actually studied independently. There are several studies that suggest the genetic contribution to smoking phenotypes differs between European and Asian ancestry- yet this is not addressed in discussion or in the analyses.

More details are now given to show that all the associated markers in both populations were independent on smoking status (Supplementary figure 1; main text line 237-245, page 7). As can be seen in Supplementary Tables 8 and 9, results are extremely similar to original results in Tables 2 and 3, indicating that adjustment by smoking makes little difference, just minor fluctuations in the p-values and ORs. In addition, the Asian replication sample (non-smokers) gave very similar results to the initial sample. This is in line with the evidence for lung cancer, where all susceptibility loci appear to be independent of smoking status with the exception of the 15q25 variant (McKay et al Nature Genetics 2017).

4. The extensive discussion of HLA grooves and structure is interesting and predicated on one tool-again one that others might not agree as the optimal. This should be discussed. Shortening this discussion might be an enhancement of the manuscript as the interpretation is quite long when no new data is presented.

We do recognize the reviewers point, and have stated the need for future functional analyses of our findings in the main text (line 360, page 11).

5. It is hard to say that a fair comparison across ethnic groups has been conducted so the title is not well supported. This reviewer strongly encourages the focus to go deeper into the European data and make this the central focus. The numbers of Asians is smaller and less compelling- small enough that it is hard to make much of a set of conclusions.

As this is the first study describing in depth a HLA association with lung cancer for both ethnicities, and as we have increased the size of the Asian dataset, we prefer to maintain the focus on both ethnicities.

Reviewers' comments:

Reviewer #1 (Remarks to the Author):

Authors performed fine mapping analysis of the MHC region in lung cancer using clinical samples from both Caucasian and Asian populations. The fine mapping analysis was performed in two population samples separately and identified different HLA associations. In Caucasians, the previously reported associations were driven by the variants within both HLA-B and –DRB1 loci, whereas the associations were driven by the variants within HLA-DQB, DRB1 and A loci. This study has advanced our understanding about the association of the HLA alleles with lung cancer risk.

1. As demonstrated before, the imputation analysis of the MHC region, particular HLA alleles and amino acids, has much worse performance in Asian populations than Caucasians. As a result, it is problematic to do the association test using the best-guess genotypes. As most of other HLA fine mapping studies, the association test should be done by incorporating the uncertainty of genotype imputation.

2. The sample size is much larger for Caucasian sample than Asians. Authors also need to discuss the impact of the difference in sample size as well as imputation performance (see above) on the comparison of the results between Caucasian and Asian populations.

3. Generally, a fine mapping analysis of the MHC region is performed by including all the HLA classic alleles, amino acid polymorphisms and SNPs so that primary driver(s) of the associations with the MHC region can be determined through stepwise conditional analyses. In the current study, the HLA classic alleles and AAs were analyzed separately, which made it a bit of harder to understand which is the primary driver(s) of the previously reported associations.

4. In the Asian samples, all the associations reported in this study did not reach genome-wide significance. This made the interpretation of the results difficult, particularly giving the fact that HLA imputation was poor in Asian samples, and the Asian samples have a very small sample size. As pointed out, rs3129860 is the top association marker within the region. Will all the HLA associations disappear after conditioning on this SNP? If this is the case, what will be reason to believe that this SNP is the “marker”, whereas HLA associations are causal? This is critical giving that the top HLA AA positions do not seem to have any functional relevance.

5. Some HLA alleles are common in both Caucasian and Asian populations, but only show association in one population. Authors need to provide the result of power calculation analysis for all the reported HLA alleles in the two populations. If the power is sufficient, but only one population showed association, does this suggest that such a HLA allele is likely to be a “marker” instead of a functional association? Giving that the sample size and imputation performance is quite different and the results of the Asian population is still questionable, it is premature to claim that the genetic risk of the MHC region on lung cancer is different between Asian and European populations.

Reviewer #2 (Remarks to the Author):

The revised manuscript carefully addresses the concerns I had and the authors have done a remarkable job at complementing their results with new Asian cohort for 8500 Asians and validated their original results and further strengthen the lack of overlap between hits among Asian and Europeans. The methods are clearer and the figures are much improved.

Reviewer #3 (Remarks to the Author):

Overall, the paper has improved but there remain several substantive issues pointed out by both the other reviewers as well. There are three major issues that require further response:

1. The authors have introduced the nonsmoking Asian GWAS in women, which, while important raises some fundamental questions not adequately addressed in the discussion- as well as the conclusions. The differences between Asian and European are expected given the genetics of HLA and the population- private nature of both alleles and distributions- so the conclusion is NO SURPRISE. So, the authors need to be more careful. Furthermore, the difference in sets between smokers and non-smokers is critical and can not be overstated.
2. Laboratory validation of notable SNPs identified by imputation must be performed in a subset. This is standard practice and surprising that this was not done-even after the first set of reviews. The daunting nature of imputation, especially in the HLA region requires further testing of a subset. It is telling that when the authors tried to use the recent Nature Genetics Han set, the performance was so poor. Many are still skeptical of the accuracy and precision of imputation- and in this regard, actual genetic data is necessary. Statistical re-imputing subsets for validation represents circular logic.
3. The notion of different roles for HLA by population is somewhat of a misnomer. Related to #1 above, the issues are confounded by smoking as to exactly what are the proposed mechanisms. Since this paper has no functional data, the focus should be on mapping and the handwaving on functional implications minimized (revised further). This is a mapping paper (still in need of validation). There are other tools- and it is surprising that they were not employed here

Reviewers' comments:

Reviewer #1 (Remarks to the Author):

Authors performed fine mapping analysis of the MHC region in lung cancer using clinical samples from both Caucasian and Asian populations. The fine mapping analysis was performed in two population samples separately and identified different HLA associations. In Caucasians, the previously reported associations were driven by the variants within both HLA-B and –DRB1 loci, whereas the associations were driven by the variants within HLA-DQB, DRB1 and A loci. This study has advanced our understanding about the association of the HLA alleles with lung cancer risk.

1. As demonstrated before, the imputation analysis of the MHC region, particular HLA alleles and amino acids, has much worse performance in Asian populations than Caucasians. As a result, it is problematic to do the association test using the best-guess genotypes. As most of other HLA fine mapping studies, the association test should be done by incorporating the uncertainty of genotype imputation.

We thank the reviewer for this comment. In fact we used genotype dosages (Methods line 79, page 3) to perform association tests for all the analysis except for the haplotype analyses for which we employed phased genotypes. We clarified this in Results (Main text; line 163, page 6) as the initial sentence was not enough accurate.

2. The sample size is much larger for Caucasian sample than Asians. Authors also need to discuss the impact of the difference in sample size as well as imputation performance (see above) on the comparison of the results between Caucasian and Asian populations.

The Asian sample size issue is included in the discussion (line 337-338 and 346-357, page 12-13). As suggested by the reviewer, we now included a more accurate comment on the imputation performance for Asians in results (line 146-153, page 6).

3. Generally, a fine mapping analysis of the MHC region is performed by including all the HLA classic alleles, amino acid polymorphisms and SNPs so that primary driver(s) of the associations with the MHC region can be determined through stepwise conditional analyses. In the current study, the HLA classic alleles and AAs were analyzed separately, which made it a bit of harder to understand which is the primary driver(s) of the previously reported associations.

We did include all the markers -HLA classic alleles, amino acid polymorphisms and SNPs- for stepwise conditional analyses. This can also be checked in the plots displayed in Figure 1 and 2 where each point corresponds to one single marker and different colors represent their type. After this step, we focused on the amino acids within the independent associated regions to select the best model fitting the data, but only after identifying the most likely independent genes-HLA alleles and/or SNPs driving the association (Methods; lines 89-97, page 3 and lines 128-138, page 4-5).

4. In the Asian samples, all the associations reported in this study did not reach genome-wide significance. This made the interpretation of the results difficult, particularly giving the fact that

HLA imputation was poor in Asian samples, and the Asian samples have a very small sample size. As pointed out, rs3129860 is the top association marker within the region. Will all the HLA associations disappear after conditioning on this SNP? If this is the case, what will be reason to believe that this SNP is the “marker”, whereas HLA associations are causal? This is critical giving that the top HLA AA positions do not seem to have any functional relevance.

Although rs3129860 was the best associated SNP in the discovery, it was also highly correlated with HLA-DQB1*0401 allele ($r^2=0.75$). The same situation happened after conditioning on it with the second best associated variant, so not much difference is expected in conditional analysis using these SNPs instead (see table below). The reason to include HLA alleles as covariates when they or their tag-SNPs appeared as the best associated markers is primarily to identify other independent effects outside HLA genes/HLA haplotypes if they exist. As the best associated SNPs were not annotated in any functional region (methods; line 89-97, page 3), it is expected that they are “markers” of the HLA allele they are correlated with. The fact that the amino acid change Ala-104 in HLA-DRB1, one of the two replicated variants in Asians, is indeed above the GWA’s threshold (combined p-value: 5.21×10^{-10}) (Table 5) and the top associated variant in the combined analyses support this analytic strategy even though is not a supported a priori by functional information.

Table A (for response to reviewer only): Top associations with adenocarcinoma of Asian ancestry using the best associated SNPs instead of their correlated HLA variants.

Conditioning on	Locus	Variant	Multivariate analysis ^a	
			OR (95%CI)	P value
none	Intergenic (Class II)	rs3129860	1.62 (1.37-1.93)	3.56×10^{-08}
	HLA_A (intronic)	rs2256919	0.75 (0.67-0.83)	1.75×10^{-07}
	HLA-DRB1	104(Ala)	1.33 (1.20-1.50)	1.96×10^{-07}
		07:01	1.62 (1.31-2.01)	5.48×10^{-06}
	HLA-DQB1	04:01	1.67 (1.35-2.05)	1.59×10^{-06}
rs3129860	HLA_G (intronic)	rs9258631	2.0 (1.56-2.55)	2.89×10^{-08}
	HLA_A (intronic)	rs2256919	0.75 (0.67-0.84)	3.36×10^{-07}
	HLA-DRB1	07:01	1.72 (1.39-2.14)	6.67×10^{-07}
		104(Ala)	1.27 (1.13-1.42)	4.10×10^{-05}
	HLA-DQB1	04:01	1.17 (0.85-1.60)	0.34
rs3129860 + rs9258631	Intergenic (Class II)	rs3129860	1.64 (0.03-7.52)	2.0×10^{-08}
	HLA_G (intronic)	rs9258631	2.0 (1.56-2.55)	2.89×10^{-08}
	HLA_A (intronic)	rs2256919	0.78 (0.65-0.92)	5.70×10^{-06}
	HLA-DRB1	104(Ala)	1.2 (1.07-1.35)	0.002
		07:01	1.36 (1.06-1.75)	0.02
HLA-DQB1	04:01	1.17 (0.85-1.61)	0.33	
rs3129860 + rs9258631 + rs2256919	Intergenic (Class II)	rs3129860	1.63 (1.37-1.93)	3.93×10^{-08}
	HLA_G (intronic)	rs9258631	1.87 (1.46-2.39)	7.40×10^{-07}
	HLA_A (intronic)	rs2256919	0.78 (0.69-0.87)	4.35×10^{-06}

HLA, human leucocyte antigen; OR, odds ratio; 95%CI, confidence interval

^a Obtained from multivariate unconditional logistic regression assuming an additive genetic model with sex and principal components as covariates

The study-wide significant threshold was $P=6 \times 10^{-6}$ (Bonferroni correction)

5. Some HLA alleles are common in both Caucasian and Asian populations, but only show association in one population. Authors need to provide the result of power calculation analysis for all the reported HLA alleles in the two populations. If the power is sufficient, but only one population showed association, does this suggest that such a HLA allele is likely to be a “marker” instead of a functional association? Given that the sample size and imputation performance is quite different and the results of the Asian population is still questionable, it is premature to claim that the genetic risk of the MHC region on lung cancer is different between Asian and European populations.

As requested by the reviewer, we provide statistical power calculations for both populations but also imputation probabilities for the specific variants in supplementary table 12 and 13 (main text; lines 212-216, page 8). While imputation was good for all the associated variants in both populations, the power was insufficient for detecting some European hits in Asians but not the opposite. This is probably because of the differences in the spectrum of antigens in the two populations that might introduce changes in which alleles might play the important role in disease susceptibility within each population, so is not surprising to find these contrasts. But as we already stated in the discussion (line 346-357, page 13-14), we cannot discard the same effects in squamous cell carcinoma in both populations. However, it seems that HLA is not playing a role in adenocarcinoma in Europeans, unless the effects are hidden in very rare variants. Given this, we agree on being more careful with the conclusion regarding the different HLA genetic background between Europeans and Asians (line 359-364, page 14).

Reviewer #2 (Remarks to the Author):

The revised manuscript carefully addresses the concerns I had and the authors have done a remarkable job at complementing their results with new Asian cohort for 8500 Asians and validated their original results and further strengthen the lack of overlap between hits among Asian and Europeans. The methods are clearer and the figures are much improved.

Reviewer #3 (Remarks to the Author):

Overall, the paper has improved but there remain several substantive issues pointed out by both the other reviewers as well. There are three major issues that require further response:

1. The authors have introduced the nonsmoking Asian GWAS in women, which, while important raises some fundamental questions not adequately addressed in the discussion- as well as the conclusions. The differences between Asian and European are expected given the genetics of HLA and the population- private nature of both alleles and distributions- so the conclusion is NO SURPRISE. So, the authors need to be more careful. Furthermore, the difference in sets between smokers and non-smokers is critical and cannot be overstated.

We understand the reviewer's concern about the differences in the smoking rates between European and Asians and the possible influence of this fact in the lack of associations overlapping between ethnicities. While we consider it beyond doubt that Asian HLA associations are independent on smoking status (Supplementary Figure 1c-f and Supplementary table 9), especially as we have replicated our initial findings in the non-smoking Asian women dataset, it is true that the proportion of non-smokers in Asian AD (469/1192; 39%) is higher than in European AD (1004/7088; 14%) and this could represent a problem if there is a dependency on smoking. However, we did not see the association in European AD non-smokers even though we had enough power to detect the effects shown in Asian AD.

We now remark on the differences between Asians and Europeans in this sense and to address this in the discussion (line 350-357, page13-14) and conclusions (line 366-371, page14).

2. Laboratory validation of notable SNPs identified by imputation must be performed in a subset. This is standard practice and surprising that this was not done-even after the first set of reviews. The daunting nature of imputation, especially in the HLA region requires further testing of a subset. It is telling that when the authors tried to use the recent Nature Genetics Han set, the performance was so poor. Many are still skeptical of the accuracy and precision of imputation- and in this regard, actual genetic data is necessary. Statistical re-imputing subsets for validation represents circular logic.

In response to this comment, we have performed a laboratory validation of the variants identified by imputation using another genotyping platform, the Affymetrix Axiom exome array (Kachuri et al. 2016, Carcinogenesis), in a subset of 5,742 individuals from the European series (Methods lines 51-70, page 2-3 and Main text lines 154-162, page 6). A separate laboratory validation was not available for the Asian populations, as discussed in lines 156-162, although we did replicate our results among never smokers in an independent series.

Regarding statistical re-imputation, we would stress that this was done using a completely different algorithm and reference panel to confirm imputation results (methods; lines 46 and 53-55, page 2).

In our analysis, the accuracy of the Asian imputation results using the initial or the Han reference panel were not comparable because the attempted imputation using the Han panel failed due to insufficient computational resources for such a large reference panel. Therefore, both common and rare variants were affected. By contrast, just the low frequency variants were compromised using the initial panel because of its modest sample size. Given the statistical power of the Asian set (Supplementary tables 12 and 13), we would argue that this scenario is not affecting the association analysis as we focused on common variants (Main text lines 149-153, page 6).

We therefore consider the concordance between imputed genotypes and their validation results from an alternative genotyping platform or re-imputation (>95% for all variants) satisfactory and results from this study already validated (Supplementary Table 10 and 11; Main text lines 160-162, page 6).

3. The notion of different roles for HLA by population is somewhat of a misnomer. Related to #1

above, the issues are confounded by smoking as to exactly what are the proposed mechanisms. Since this paper has no functional data, the focus should be on mapping and the handwaving on functional implications minimized (revised further). This is a mapping paper (still in need of validation). There are other tools- and it is surprising that they were not employed here.

We are presenting this paper as a fine-mapping analysis and only secondarily have we pointed out possible functional implications using different tools and specifying that this should be followed up by functional analysis (line 321, page 12). We think that adding possible perspectives on the results based on well-known and widely used functional annotation tools (Methods; line 89-97, page 3) is interesting and we prefer to maintain this in the discussion although we have reduced it by avoiding specific conclusions on the putative role of the associated variants.

REVIEWERS' COMMENTS:

Reviewer #1 (Remarks to the Author):

My concerns/questions have been addressed.

Reviewer #3 (Remarks to the Author):

Thank you for your careful response to the last set of comments.

The paper is improved and more balanced in discussing the findings.

It is a shame that further validation genotyping was not done in a subset of the Asian cases/controls. This was strongly preferred and really the standard in most high quality GWAS papers.